Resource

# Twisting the theory on the origin of human umbilical cord coiling featuring monozygotic twins

Pia Todtenhaupt[1], Thomas B Kuipers[1,2], Kyra L Dijkstra[3], Lenard M Voortman[4], Laura A Franken[1], Jip A Spekman[5], Thomas H Jonkman[1], Sophie G Groene[5], Arno AW Roest[6], Monique C Haak[7], EJoanne T Verweij[7], Melissa van Pel[8,9], Enrico Lopriore[5], Bastiaan T Heijmans[1], Lotte E van der Meeren[3,10]

**The human umbilical cord (hUC) is the lifeline that connects the fetus to the mother. Hypercoiling of the hUC is associated with pre- and perinatal morbidity and mortality. We investigated the origin of hUC hypercoiling using state-of-the-art imaging and omics approaches. Macroscopic inspection of the hUC revealed the helices to originate from the arteries rather than other components of the hUC. Digital reconstruction of the hUC arteries showed the dynamic alignment of two layers of muscle fibers in the tunica media aligning in opposing directions. We observed that genetically identical twins can be discordant for hUC coiling, excluding genetic, many environmental, and parental origins of hUC coiling. Comparing the transcriptomic and DNA methylation profile of the hUC arteries of four twin pairs with discordant cord coiling, we detected 28 differentially expressed genes, but no differentially methylated CpGs. These genes play a role in vascular development, cell–cell interaction, and axis formation and may account for the increased number of hUC helices. When combined, our results provide a novel framework to understand the origin of hUC helices in fetal development.**

## Introduction

The genesis of the umbilical cord helices remains an elusive phenomenon in fetal development. The human umbilical cord (hUC) supplies nutrients and oxygen to the fetus and is indispensable for the adequate development of the fetus. The hUC develops 4–6 wk post-conception. It is mainly composed of the connective tissue, Wharton's jelly, which supports and protects two umbilical arteries, and one umbilical vein (1). The umbilical vein transports nutrients and oxygen from the placenta toward the fetus, whereas in the umbilical arteries, the deoxygenated blood with waste materials flows from the fetus to the placenta (2). The presence of helices in the hUC was noted as early as 1,521 (2). A helix is a twisted, three-dimensional spiral shape. Over the years, synonyms such as coils, spirals, and turns were introduced and used as analogues. Helices in the hUC arise at an early stage of development. In week 7 of gestation, ~95% of the fetuses have developed the helices, whereas ~5% of the hUCs remain un- or hypocoiled (1, 3, 4). Even though the hUC elongates, the number of helices does not change for the remaining gestation (4). Thus far, the origin of the helices themselves and the intensity of coiling remain unclear.

Insight into the origin of the helices is important because hypercoiling is strongly associated with adverse pre- and postnatal clinical outcomes (5, 6, 7). In 1994, Strong et al proposed a standardized metric to quantify hUC helices, the umbilical coiling index (UCI), which is defined as the number of helices per centimeter of the hUC and is irrespective of the helix direction (8, 9, 10, 11). In uncomplicated pregnancies, the average UCI lies around 0.17 helices/cm with the 10th and 90th centiles at 0.07 and 0.3 helices/cm, respectively, and the hUC is categorized as *normocoiled* (1, 12). When the number of helices exceeds 3 helices per 10 cm, the hUC is defined as *hypercoiled*.

Numerous studies linked an abnormal hUC coiling intensity to pre- and perinatal morbidity and mortality (5, 6, 7, 13, 14). Hypercoiling is associated with fetal growth restriction and fetal distress resulting in (planned) premature delivery, a decreased Apgar score at 5 min, or even fetal demise (6, 7). Hypercoiling can also lead to umbilical vascular compression and reduced fetal blood flow, which in turn may result in parenchymal abnormalities such as fetal thrombosis in the hUC vessels, chorionic plate, and (stem) villi, creating a higher risk of prenatal and perinatal morbidity and mortality (15). These findings underline

---

[1]Molecular Epidemiology, Department of Biomedical Data Sciences, Leiden University Medical Center, Leiden, Netherlands    [2]Sequencing Analysis Support Core, Department of Biomedical Data Sciences, Leiden University Medical Center, Leiden, Netherlands    [3]Department of Pathology, Leiden University Medical Center, Leiden, Netherlands    [4]Department of Cell and Chemical Biology, Leiden University Medical Center, Leiden, Netherlands    [5]Neonatology, Willem-Alexander Children's Hospital, Department of Pediatrics, Leiden University Medical Center, Leiden, Netherlands    [6]Pediatric Cardiology, Willem-Alexander Children's Hospital, Department of Pediatrics, Leiden University Medical Center, Leiden, Netherlands    [7]Department of Obstetrics, Division of Fetal Therapy, Leiden University Medical Center, Leiden, Netherlands    [8]NecstGen, Leiden, Netherlands    [9]Department of Internal Medicine, Leiden University Medical Center, Leiden, Netherlands    [10]Department of Pathology, Erasmus Medical Center, Rotterdam, Netherlands

Correspondence: L.vandermeeren@lumc.nl

    

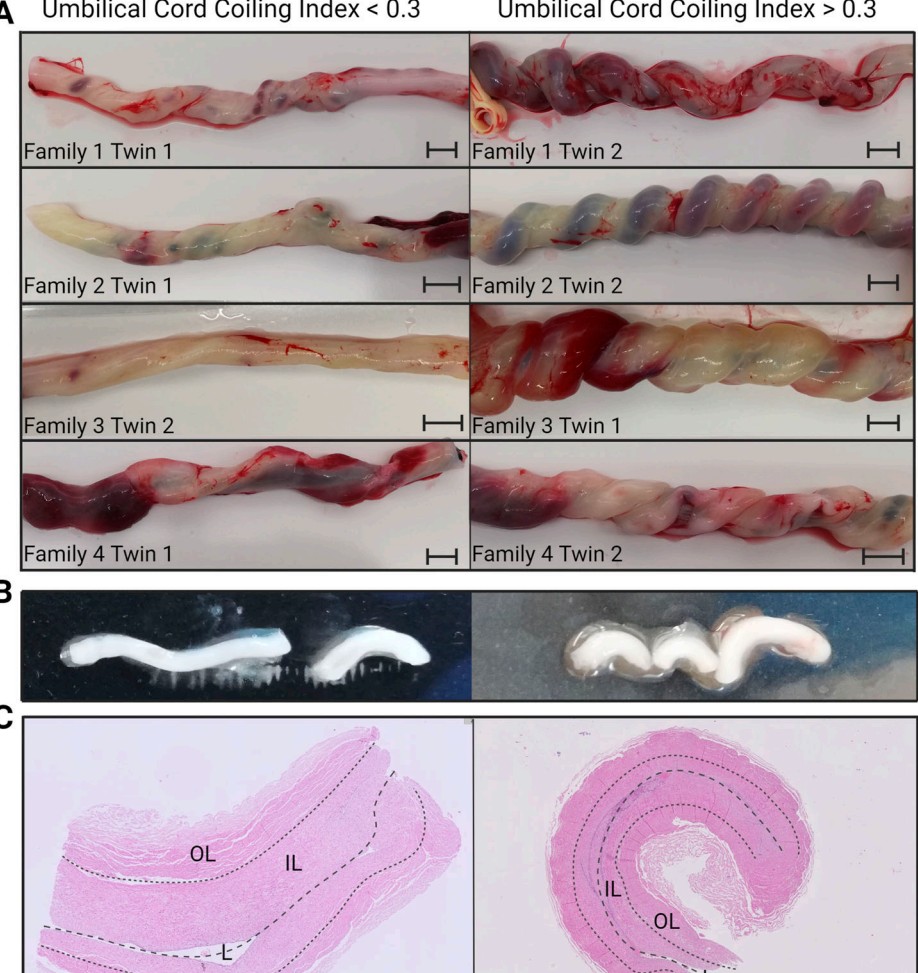

**A** Umbilical Cord Coiling Index < 0.3    Umbilical Cord Coiling Index > 0.3

Family 1 Twin 1 — Family 1 Twin 2 —
Family 2 Twin 1 — Family 2 Twin 2 —
Family 3 Twin 2 — Family 3 Twin 1 —
Family 4 Twin 1 — Family 4 Twin 2 —

**B**

**C** OL IL L    IL OL L

**Figure 1. Overview of the included umbilical cords.**
**(A)** Overview of the included umbilical cords and coiling. Hypercoiling is defined by an umbilical cord coiling index > 0.3; genetically, controls have an umbilical coiling index < 0.3. The scale bar indicates 1 cm. **(B)** Example of the isolated arteries from each condition; hypercoiled (right), control (left). **(C)** Hematoxylin and eosin staining of longitudinal cross sections of hyper- and normocoiled umbilical arteries. The umbilical arteries consist of two muscle fiber layers, the outer layer (OL) and the inner layer (IL). The middle dashed line indicates the course of the lumen (L). Left: normal coiled umbilical artery; the scale bar indicates 500 $\mu$M. Right: hypercoiled umbilical artery; the scale bar indicates 1 mm.
Source data are available for this figure.

the urgency to further explore the origin of the helices and thereby the adverse pregnancy outcomes. However, research into the origin of coiling stagnated in the 90s of the last century after several hypotheses on the development of helices were introduced and subsequently shelved or invalidated (1). State-of-the-art imaging and molecular approaches, which were unavailable during the initial wave of studies, have the potential to alter this situation.

In this study, we investigated the histology of hypercoiled hUCs using digital reconstruction and investigated the basis of the hUC helices in newborn identical twin pairs discordant for their hUC coiling index. Identical twin pairs share their genome, sex, parental factors, and most of the intrauterine environmental factors, minimizing confounding when comparing hyper- and normocoiled hUCs. To define the molecular signature and origin of hUC helices, we generated transcriptomic data using RNA-seq and genome-wide epigenetic data using DNA methylation array profiling >700 1000 locations throughout the genome.

# Results

## Clinical and macroscopic hUC characteristics of MZ twin pairs discordant for coiling

Within a period of 6 mo, we identified four MZ twin pairs with a macroscopically visible difference in coiling index further referred to as a discordant hUC (Fig 1A). The individuals of the MZ twin pair with a hypercoiled hUC (classified as cases, UCI between 0.3 and 0.45), whereas the hUC of the other sibling was normocoiled (classified as controls, UCI between 0.09 and 0.29). The length of the hUC was comparable between the individuals of each of the four twin pairs and was within normal limits (30–70 cm) (9, 16). The helical pattern was evenly distributed throughout the hUC. We did not observe increased morbidity in the twin with the hypercoiled umbilical cord compared with its co-twin. The twin pairs were born at a gestational age ranging from 29 to 36 wk, with an equal distribution of male (2) and female (2) (Table S1). The arteries were retrieved from four MZ twin pairs, in total eight hUCs. Remarkably,

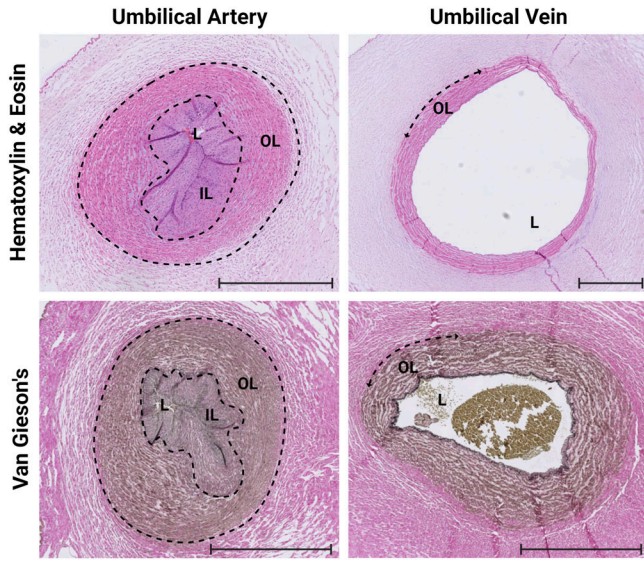

**Umbilical Artery** | **Umbilical Vein**

Hematoxylin & Eosin

Van Gieson's

**Figure 2. Umbilical cord artery and vein muscle fiber alignment.**
Hematoxylin and eosin and Van Gieson's staining of the artery and vein derived from a hypercoiled umbilical cord. The umbilical artery wall (intima–media) consists of two layers of muscle fibers, the outer layer (OL) and the inner layer (IL), aligned in crossing directions. The wall of the umbilical vein consists of one outer layer, a circular muscle fiber in one direction. L, lumen. The scale bar indicates 1 mm.
Source data are available for this figure.

the helices were retained in each of the two arteries upon removal. The remaining hUCs containing Wharton's jelly and the umbilical vein exhibit no inherent coiling, suggesting that the source of umbilical cord coiling originates solely from the umbilical cord arteries (Fig 1B and C).

## Histological evaluation of the arterial muscle layers

Given that the arteries seem to play an essential role in coiling, we performed histological evaluation of the umbilical arteries. H&E-stained cross sections of the isolated arteries revealed three anatomical structures from the outer layer, the tunica adventitia, the tunica media, and the tunica intima with an epithelium layer on the luminal side. Furthermore, we observed within the tunica media of the umbilical artery two distinct muscle layers aligned in opposite directions, whereas all other arteries in the human body are known to be composed of a single muscle cell layer (Fig 2). As the H&E and Van Gieson's stainings were not able to quantify and visualize the fibers in multiple directions, we constructed a serial animation of 100 consecutive H&E slides of a hypercoiled artery to assess and evaluate the dynamic alignment of both the inner and outer muscle layers. With this approach, we observed the movement of both muscle layers in opposite crossing directions (Video 1). Contrarily, the wall of the vein originating from a hypercoiled umbilical cord consists of a single layer of circular muscle fibers aligned solely in one direction (Fig 2). Hence, it is plausible that the number of helices per cm may be attributed to the difference in muscle fiber alignment of the two separate layers of the arterial muscle wall.

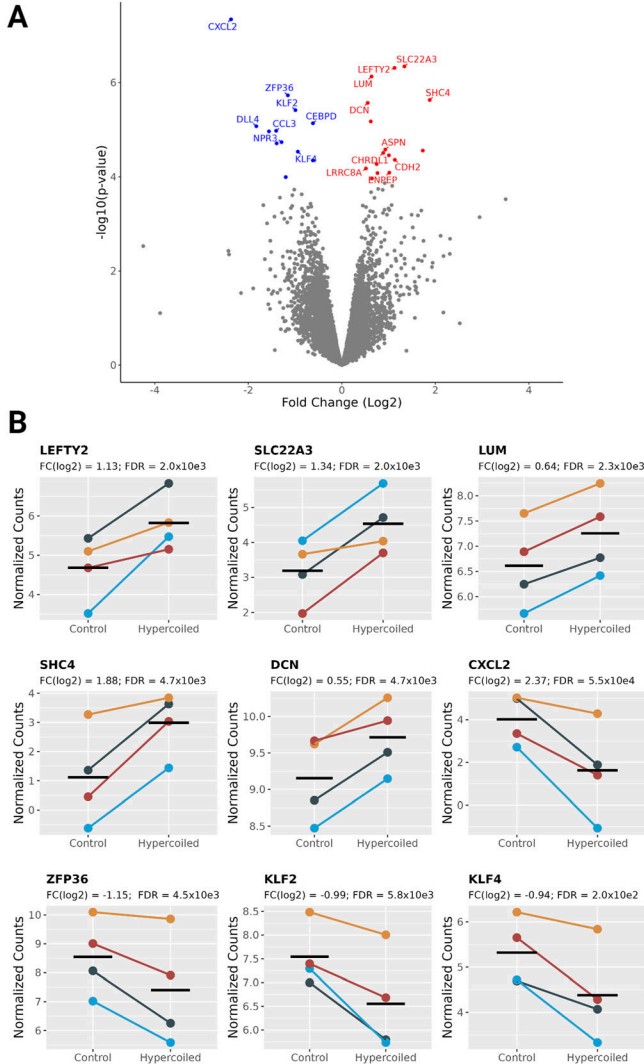

**Figure 3. Overview of the differential gene expression between hypercoiled and control arteries.**
**(A)** Volcano plot displaying a gene expression fold change between hypercoiled and normocoiled arteries. Genes that are up-regulated ($P_{FDR}$ < 0.05) are shown in red. Genes down-regulated are shown in blue. Non-significantly changed genes ($P_{FDR}$ > 0.05) are displayed in gray. **(B)** Normalized read counts of the top eight most significantly differentially expressed genes and *KLF4*. Twin pairs of family 1–4 are depicted in dark green, orange, light blue, and dark red, respectively.

## Hypercoiled umbilical cord arteries display a distinct transcriptomic profile

To gain insight into the molecular distinctions in the arterial muscle tissue involved in the increased formation of helices, transcriptomic data of the hUC arteries (n = 8) were generated. Comparing the gene expression of hypercoiled and normocoiled hUC arteries, 28 genes were significantly differentially expressed ($P_{FDR}$ > 0.05; Fig 3A and Table S2). In hypercoiled arteries, 16 of the 28 genes were up-regulated ($\log_2$ fold change: 0.5–1.9), whereas 12 of the 28 genes were down-regulated ($\log_2$ fold change: –0.6 to –2.4) as compared to normocoiled arteries (Fig 3B).

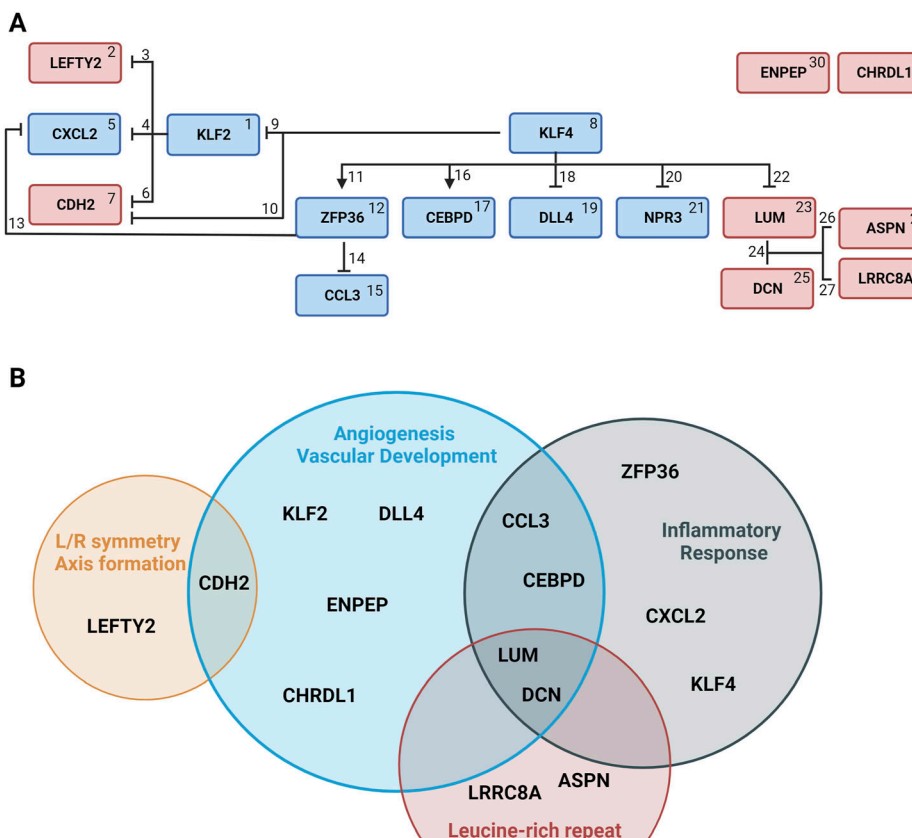

**A**

**B**

**Figure 4. Schematic overview of the function and interaction of differentially expressed genes between hypercoiled and control umbilical arteries.**
**(A)** Interaction of a selected differentially expressed gene set between hypercoiled/case and normocoiled/control umbilical arteries. Numbers assigned to the genes and interactions correspond to Table 1, containing a description of the function and interaction of each gene. Genes that are overexpressed are depicted in a red box ($P_{FDR} < 0.05$). Genes that are down-regulated are depicted in a blue box. Arrows indicate a positive regulatory correlation found in previous studies. The inhibitory arrow indicates a negative regulatory correlation found in previous studies. **(B)** Most frequently occurring functional terms in the manual exploration of the differentially expressed genes including genes related to the term. Several genes are shared between different functional terms. The size of the cycles corresponds to the number of genes related to the functional term. This list is a selection of identified functions and interactions and does not contain a complete list of all published functions and interactions.

## Pathway enrichment analysis of genes associated with hypercoiling

To explore the biological processes represented by the differentially expressed genes in normo- and hypercoiled hUC arterial tissues, a gene set enrichment analysis was performed. Enrichment was observed for *small leucine-rich proteoglycan molecules*, *cell differentiation*, and *response to laminar shear stress* (Fig S1A). Most of the enriched processes (13 of 23) contained the up-regulated transcription factors *KLF2* and/or *KLF4* (Fig S1B).

As the function of the genes showing transcriptomic differences between hypercoiled and normocoiled hUC arterial tissues may not be fully captured by processes as they were previously specified in databases, we manually annotated the function of differentially expressed genes using published work. We found that 9 of 28 genes were involved in vessel formation (*KLF2, CDH2, CCL3, CEBPD, DLL4, LUM, DCN, ENPEP,* and *CHRDL1*), 6 of 28 were involved in inflammatory responses (*CXCL2, KLF4, ZFP36, CCL3, CEBPD,* and *LUM*), four were leucine-rich repeat–containing proteins (*LUM, DCN, ASPN,* and *LRRC8A*), and two genes determine axis formation by guiding the left–right patterning (*LEFTY2* and *CDH2*) (Fig 4A and B and Table 1). Collectively, our functional annotation links the transcriptomic profile of hypercoiled hUC arteries to aspects of processes influencing the alignment of muscle cells, thereby contributing to the hUC coiling index.

## Gene expression changes in hypercoiled arteries are not present in hUC-MSCs

To further probe whether the observed changes in gene expression are restricted to the hUC arterial tissue or reside more globally, we inspected the gene expression profile of a cell type isolated from Wharton's jelly of the hUC, human umbilical cord mesenchymal stromal cells (hUC-MSCs). No genes were found to be differentially expressed, when comparing the transcriptomic profile in hUC-MSCs of the four twin pairs discordant for coiling. Seven of the 28 genes differentially expressed in the hUC arterial tissue were not expressed in hUC-MSCs, and the remaining 21 genes are not among the transcripts on the lower end of detected *P*-values ($0.92 > P_{FDR} > 0.07$; Fig S2). This may indicate that the observed transcriptomic profile is specific to the isolated hUC arterial muscle fiber tissue.

## Arterial DNA methylation signature remains unchanged upon hypercoiling

To investigate whether epigenetic changes underlie the discordance in gene expression between hyper- and normocoiled umbilical arteries, a genome-wide DNA methylation profile of the hUC arterial tissue was established for each individual of the four MZ twin pairs discordant for coiling (729,706 CpGs). None of the measured CpG sites displayed a differential methylation level when

**Table 1.   Function and interactions of differentially expressed genes between hypercoiled and control umbilical arteries.**

| | |
|---|---|
| 1 | KLF2 is a key regulator of multiple endothelial functions, maintaining a healthy endothelium (17, 18). KLF2-null mouse embryos show incomplete vascular maturation through insufficient migration of mural cells to the arterial wall (19). Although in the absence of KLF2, full vessel maturation is impaired, primary vasculogenesis is not affected (20). Shear stress–medicated actin cytoskeleton remodeling is KLF2-dependent in endothelial cells. KLF2-induced actin shear fibers facilitate endothelial cell alignment in the direction of the flow (21). |
| 2 | Transforming growth factor-β–related factor, LEFTY2, acts as an asymmetric signaling molecule by antagonizing the duration and site of the left-side determinant NODAL (22). |
| 3 | Lentiviral-mediated overexpression of KLF2 in human umbilical vein endothelial cells results in down-regulation of LEFTY2 (21). |
| 4 | IL-1β–mediated increase in CXCL2 is muted by KLF2 in human umbilical vein endothelial cells (18). |
| 5 | CXCL2 is a pro-inflammatory chemokine playing a role in the development of cardiovascular diseases (18, 23). The overexpression of CXCL2 can lead to vascular endothelial cell damage in blood vessels through the activation of neutrophils. Furthermore, CXCL2 plays a role in reorganization of the cytoskeleton, cell migration, adhesion, and immune response and is associated with atherosclerosis, diabetes, obesity, and myocardial infarction (23). |
| 6 | Lentiviral-mediated overexpression of KLF2 in human umbilical vein endothelial cells results in down-regulation of CDH2 (21). |
| 7 | CDH2 (N-cadherin) promotes angiogenesis and enhances the stability of blood vessels by acting as a cell–cell adhesion molecule to enhance the interaction between endothelial cells and mural cells (24, 25). Furthermore, CDH2 plays a role in establishing the left–right axis during development, independent of LEFTY and NODAL. Blocking CDH2 function during development of chicken embryos randomizes heart looping (26). |
| 8 | KLF4 orchestrates transcriptional programs as a key regulator of multiple endothelial functions ensuring an anti-inflammatory and antithrombotic endothelial phenotype (together with KLF2) (17, 27). |
| 9 | In endothelial cells, more than 40% of the genes regulated by KLF2 are similarly regulated by KLF4 (27). The adenoviral-mediated overexpression of KLF4 in endothelial cells results in down-regulation of KLF2 (27). |
| 10 | Adenoviral-mediated overexpression of KLF4 in endothelial cells results in down-regulation of CDH2 (27). |
| 11 | Adenoviral-mediated overexpression of KLF4 in endothelial cells results in up-regulation of ZFP36 (27). |
| 12 | ZFP36 (TTP) is an RNA-binding protein, guiding pro-inflammatory cytokine mRNA for degradation or regulating their translation and thereby decreasing pro-inflammatory responses (28, 29). |
| 13 | CXCL2 was identified as a ZFP36 target in mouse macrophages (29). |
| 14 | CCL3 was identified as a ZFP36 target in mouse macrophages and fibroblasts (29) |

**Table 1.   Continued**

| | |
|---|---|
| 15 | CCL3 or macrophage inflammatory protein α (MIP-1α) is a pro-inflammatory chemokine (30). It has been identified as an angiogenic factor in osteosarcoma, where it plays a role in inducing endothelial progenitor cell migration and tube formation (31). |
| 16 | Adenoviral-mediated overexpression of KLF4 in endothelial cells results in up-regulation of CEBPD (27). |
| 17 | CEBPD is part of the CCAAT/enhancer-binding protein family and plays a role in differentiation, metabolism, immune response, and inflammatory disease. CEBPD promotes proliferation, migration, and in vitro tube formation of human umbilical vein endothelial cells and thereby contributes to angiogenesis (32). |
| 18 | DLL4 levels were decreased upon KLF4 overexpression in the mouse retinal angiogenesis model (33). |
| 19 | DLL4 was shown to be predominantly expressed in vascular endothelium and at active sites of angiogenesis (34, 35). In the umbilical cord, DLL4 is solely expressed in the umbilical arteries but not in the umbilical vein (34). In mice, deletion of DLL4 results in abnormal artery development and defective arterial branching (36). |
| 20 | NPR3 was shown to be down-regulated upon KLF4 overexpression in a transfected colon cancer cell line (37). |
| 21 | NPR3/NPR-c functions as a clearance receptor for natriuretic peptides, thereby reducing blood pressure (38). NPR3 is among others expressed in endothelial cells, vascular smooth muscle cells, and endocardial cells (39). |
| 22 | LUM was shown to be down-regulated upon KLF4 overexpression in a transfected colon cancer cell line (37). |
| 23 | LUM is a small leucine-rich proteoglycan that plays a key role in the regulation of the stromal collagen matrix. LUM has been associated with cell migration and proliferation during embryonic development and inflammatory responses (40). Lumican was also shown to antagonize angiogenesis by inhibiting endothelial cell angiogenic sprouting and invasion (41). |
| 24 | DCN and LUM are both members of the family of small leucine-rich proteoglycans and share multiple functions related to the maintenance of tissue homeostasis, and have both been identified to inhibit angiogenesis (40, 41). |
| 25 | DCN is a small leucine-rich proteoglycan (40). DCN is a structural component of the extracellular matrix and acts as a pro-angiogenic factor by supporting endothelial cell adhesion to type I collagen and α1β2-integrin. On the contrary, DCN has been shown to antagonize angiogenesis in the context of tumorigenesis-associated angiogenesis and inflammatory processes (42, 43). |
| 26 | ASPN is a member of the leucine-rich repeat protein family like DCN and LUM. ASPN is similar and closely related to DCN (44). |
| 27 | LRRC8A is like ASPN, LUM, and DCN a leucine-rich repeat protein (45). |
| 28 | ASPN is a member of the leucine-rich repeat protein family that is expressed among others in the aorta and tissues with large abundance of smooth muscle cells (44). ASPN is an extracellular matrix protein, can bind directly to type I collagen, and contributes to collagen fibrillogenesis (46). |

| | |
|---|---|
| **29** | LRRC8A is a leucine-rich repeat–containing protein and forms an essential component of the volume-regulated anion channel (45). Down-regulation of LRRC8A decreases cerebrovascular smooth muscle cell proliferation (47). |
| **30** | ENPEP encodes for aminopeptidase, a membrane-associated protease. It is up-regulated in pericytes of tumor blood vessels. Aminopeptidase-binding peptides can inhibit its proteolytic function, which in turn affects endothelial cell function and angiogenesis (48). |
| **31** | CHRDL-1 is a bone morphogenetic protein (BMP) antagonist and is suggested to play a role in retinal angiogenesis by modulating BMP-4 actions on endothelial cells (49). In addition, CHRDL-1 was shown to inhibit proliferation and migration of amniotic fluid–derived mesenchymal stromal cells (50). |

Description of the function and interaction of differentially expressed genes between hypercoiled and control umbilical arteries. Assigned numbers in the first column correspond to Fig 3A. This list is a selection of found functions and interactions and does not contain a complete list of published functions and interactions.

comparing hypercoiled and normocoiled hUC arteries (Table S3 and Fig S3A and B). The 10 CpGs with the lowest $P$-value signifying the largest difference in the methylation level from the comparison ($P_{FDR} > 0.68$) mapped to the body of genes that are involved in a heterogeneous set of biological processes (Table 2). The CpGs located in proximity to the genes differentially expressed in the hUC arterial tissue showed no indication of differential methylation (Fig S4). Taken together, the observed differential gene expression between hypercoiled and normocoiled umbilical arteries did not coincide with differences in the methylome.

# Discussion

Several hypotheses regarding the origin of cord coiling have thus far been proposed. Yet, none of these are validated or widely recognized as a plausible explanation for the origin of hUC coiling (1). However, using state-of-the-art technologies including arterial digital reconstruction and transcriptomic analysis to study cord coiling in our unique monozygotic twin cohort, we can now challenge and refute previous explanations of the origin of cord coiling. We were able to visualize the distinct arrangement of the two muscle layers in the tunica media in the umbilical cord arteries and identify genes that are pivotal in the formation of umbilical cord helices.

The most commonly invoked explanation for hUC coiling is that the number of helices would depend on the amount of fetal movement including active and passive torsions (2). However, this is implausible because the helices develop at 7 wk of gestational age, before the ability of the fetus to actively move (2). Also, this explanation would predict that coiling may change during pregnancy, and this has never been documented. In addition, excessive growth is unlikely to underlie hypercoiling, given extensively long umbilical cords (ELUC) were associated with a variety of placental pathologies

such as true knots but not with hypercoiling in a previously conducted study (16). Similarly, a hemodynamic origin can be refuted, as in the first trimester before week 7 when helices are being formed, the blood flow is limited and insufficient to induce helices (51, 52). Around week 11, the blood flow velocity becomes measurable within the umbilical cord, and this occurs after the helices are already present in the umbilical cord (51, 52).

Our macroscopic inspection revealed that the coiling phenotype is provoked by the coiling of the arteries as hUC Wharton's jelly including the vein lost its distinctive coiled characteristics after dissecting out the artery. This observation confirms earlier research describing that the artery is fundamental to hUC coiling (2). This has led to various hypotheses on the origin of cord coiling that involve differences in muscle arrangement in the intima–media of the arteries. One hypothesis states that a small bundle of extra muscle fibers at the side of the arteries, the Roach muscle, may be responsible for cord coiling. However, the presence of the Roach muscle is not the key determinant for hUC coiling as the Roach muscle can also be present in both normo- and hypocoiled cords, and on the contrary, it can be absent in hypercoiled cords (53). Using umbilical artery digital reconstruction of consecutively cut serial slides, we found that hUC coiling most likely originates from the two separate muscle cell layers within the arterial muscle wall, the tunica media, where muscle fibers are aligned in opposite directions.

The composition of the muscle layers of umbilical cord arteries is fundamentally different than arteries found in other parts of the human body (54, 55). All other arteries, including the aorta, are histologically composed of one circular muscle layer (tunica media) with fibers aligned in a uniform circular direction. The umbilical artery is an exception and is the only artery that forms helices and consists of two muscle layers in the tunica media. Interestingly, when comparing the hUC to those of other placental mammals (Placentalia), macroscopic and histological properties support our hypothesis that helices are formed by the structural composition of the artery's muscle cell layers. The UCs of odd-toed ungulates (Perissodactyla) such as horses and alpacas are macroscopically coiled and consist histologically of two muscle layers aligned in opposite directions, similar to human umbilical cords (56, 57, 58, 59). In contrast, the UCs of even-toed ungulates (Artiodactyla) and aquatic mammals (Sirenia) such as buffaloes and Amazonian manatees, respectively, are macroscopically uncoiled and contain arteries, histologically composed of a single muscle layer (60, 61). These observations strengthen our conclusion that coiling can occur if two muscle cell layers are present in the artery that are aligned in opposite directions to form helices.

To gain a deeper understanding of the molecular basis of hUC helices, we generated a transcriptomic and DNA methylation profile of hUC arteries obtained from MZ twin pairs discordant for cord coiling. The MZ twin pairs share their genome, parental factors, and most of the intrauterine environmental conditions during fetal development. Nevertheless, even with these shared characteristics the twin pairs were distinctly discordant for the number of helices. Therefore, genetic, most environmental, and parental influences are unlikely to be the dominant determinants of cord coiling. We observed distinct differences in gene expression within these discordant twin pairs. Specifically, we identified 28 differentially

**Table 2.  Gene annotation of the 10 most differentially methylated CpGs between hypercoiled and control umbilical arteries.**

| | | Location GRCh38/hg38 | Annotated gene | Gene function | P-value |
|---|---|---|---|---|---|
| cg23549367 | ↓ | chr3 190002687-190002689 | P3H2 | Prolyl 3-hydroxylase plays a critical role in post-translational modification of fibril-forming collagens[i]. | $1.02 \times 10^{-06}$ |
| cg03055894 | ↓ | chr4 78558508-78558510 | ANXA3 | Annexin A3 is a phospholipid-binding protein and has been identified to play a role in inflammatory response and tumorigenesis[i]. | $1.86 \times 10^{-06}$ |
| cg13710662 | ↓ | chr7 151199402-151199404 | IQCA1L | IQ motif–containing AAA domain 1 like was predicted to enable ATP-binding activity[i]. | $5.64 \times 10^{-06}$ |
| cg05346527 | ↓ | chr17 50465160-50465162 | ACSF2 | Acyl-CoA synthetase family member 2 is located in the mitochondrial matrix and is involved in fatty acid metabolic processes | $6.44 \times 10^{-06}$ |
| | | | CHAD | Chondroadherin is a leucine-rich cartilage matrix protein that plays a role in the adhesion of chondrocytes[i]. | |
| cg14287565 | ↑ | chr9 124868409-124868411 | ARPC5L | Actin-related protein 2/3 complex subunit 5 like is involved in actin filament–binding activity, actin nucleation, and cell migration[i]. | $9.82 \times 10^{-06}$ |
| cg17672209 | ↓ | chr5 134521153-134521155 | — | CpG is located in an intergenic region. | $1.17 \times 10^{-05}$ |
| cg00525772 | ↑ | chr15 45715269-45715271 | AC068722.1 | AC068722.1 was identified as a long non-coding RNA (ENSG00000259200)[ii]. | $1.49 \times 10^{-05}$ |
| cg25367332 | ↑ | chr11 | AP001007.1 | AP001007.1 was identified as a long non-coding RNA (ENSG00000254932)[ii]. | $1.74 \times 10^{-05}$ |
| | | 125265978-125265980 | PKNOX2 | PBX/knotted 1 homeobox 2 is a transcription factor and plays a role in cell proliferation and differentiation[i]. | |
| cg26882909 | ↓ | chr17 37791817-37791819 | — | CpG is located in an intergenic region. | $1.81 \times 10^{-05}$ |
| cg20712631 | ↓ | chr5 172647061-172647063 | NEURL1B | Neuralized E3 ubiquitin protein ligase 1B is located in actin cytoskeleton and cytosol and involved in ubiquitin-dependent endocytosis[i]. | $2.02 \times 10^{-05}$ |

Gene annotation of the 10 most differentially methylated CpGs between hypercoiled and control umbilical arteries. Genomic location, annotated gene, gene function, and P-value are displayed. All annotated CpGs are located in the gene body. Gene region is retrieved from the UCSC Genome Browser on Human (GRCh38/hg38). Gene function is retrieved from www.genecards.org[i] or www.ensemble.org[ii]. nc indicates that the CpG is located in a non-coding region. ↑ indicates an increase in methylation in hypercoiled umbilical arteries, and ↓ indicates a decrease in the methylation level in hypercoiled umbilical arteries.

expressed genes and annotation of the function of these genes implicated a role of vascular development, inflammatory response, extracellular matrix, cell–cell adhesion, polarity, and axis formation including left–right (a)symmetry in the origin of cord coiling.

Zooming in on the differentially expressed genes, 12 of the 28 differentially expressed genes were shown to play a role in vascular development, cell–cell adhesion, polarity, and axis formation. Among the differentially up-regulated genes in hypercoiled arteries were two major transcription factors, *KLF2* and *KLF4*, playing a key role in vascular maturation and blood vessel function. Furthermore, we observed the down-regulation of *DLL4*, a gene expressed in umbilical arteries but not the umbilical vein (34). Deletion of *DLL4* has been associated with abnormal and decreased arterial development in mice (36). We also detected up-regulation of *CDH2* (N-cadherin) in hypercoiled arteries. *CDH2* is known to promote angiogenesis and stabilize blood vessels by acting as a cell–cell adhesion molecule to enhance the interaction between endothelial and mesenchymal cells (24, 25), potentially playing a role in retaining the hypercoiled state of the UCs. Genes encoding for leucine-rich repeat–containing proteins *LUM*, *ASPN*, and *DCN* were found to be up-regulated. Although *DCN* (42, 43) and *ASPN* (44, 45) can bind to type I collagen to support cell adhesion, *LUM* (40) plays a key role in regulating the stromal collagen matrix. Also, *LEFTY2* (26)

and *CDH2* (22), genes encoding for asymmetrical signaling molecules, were down-regulated in hypercoiled arteries. *CDH2* was found to play a role in establishing the left–right symmetry during development independent of *LEFTY* and *NODAL* (62).

The genes associated with cell–cell adhesion and vascular development as mentioned earlier are essential for maintaining the structural characteristics of the muscle wall. Luis Martinez-Lemus and Gasser et al extensively described the normal appearance and collagen structure of the arterial tunica media in the human body (63, 64). Each smooth muscle cell is surrounded by a basement membrane, cytoskeleton, and intracellular connections such as integrins, collagenous fibrils, and cadherins (65). Vascular smooth muscle cells of arteries in the human body are typically oriented along the longitudinal axis. The collagen fibers in the tunica media are arranged in a helical structure (64). Muscle fiber alignment of a smooth muscle cell can vary with a maximum variation of 20° (63). This variance may be emphasized by the presence of a double muscle layer of which both layers are aligned in opposite directions resulting in a helix. The observed up-regulation of type I collagen–binding *DCN* and *ASPN*, stromal matrix component *LUM*, and polarity-associated *CDH2* (*N-cadherin*) and *LEFTY2* may facilitate structural adaptation in the artery, promoting its helical configuration.

Interestingly, blocking the function of *CDH2* with an antibody led to randomized heart looping in the chicken embryo (22), a process relying on muscle fiber alignment in crossing direction, similar to the process we identified to underlie the formation of hUC helices. *CDH2* and the association with abnormal heart development are linked to the helical appearance and spatial distribution of overlapping superficial and deep myocardial fibers of the ventricular heart with concentric contraction of the myocardium (66, 67). The up-regulation of *CDH2* in hypercoiled hUCs may give an increased muscle fiber alignment, contributing to and maintaining the increased coiling.

Lastly, our analysis revealed six genes involved in the inflammatory response that were found to be differentially regulated, a process known to be closely intertwined with thrombotic disorders. *KLF2* and *KLF4* were shown to uphold an antithrombotic effect in the vessel wall (17), whereas *ZFP36* reduces an inflammatory response by guiding mRNA for degradation (28, 29). Yet, we also observed the down-regulation of the pro-inflammatory *CXCL2*. We hypothesize that observed down-regulation of the anti-inflammatory genes may be the result of the increased thrombotic risk in hypercoiled arteries. Fetal hUC vascular obstruction has been shown to be an effector of the adverse pregnancy outcomes such as stillbirth in case of hypercoiling (15).

Epigenetic marks, together with transcription factors, largely control the regulation of gene expression. DNA methylation is the most extensively studied epigenetic mark in population cohorts contributing to the regulation of gene expression. We compared the DNA methylation profile of hypercoiled and normocoiled hUC arteries. Nevertheless, no difference in DNA methylation was observed. The expected differences in the methylation level are rather small as compared to gene expression differences, and the correction for multiple testing is stringent because of the large number of tested loci. Therefore, extending this comparison to a larger cohort may increase sensitivity to detect potential differential gene regulation in hUC arteries. Another important factor is that gene expression regulation relies on a highly complex interplay of various epigenetic mechanisms and transcription factors. Therefore, it may also be the case that differences have manifested at another level of the epigenome such as in the binding of histones or other regulatory proteins, thereby affecting DNA accessibility to the transcriptional machinery.

The observed differences in gene expression between hypercoiled and normocoiled hUC arteries likely originated and manifested during early development. This is supported by the fact that the number of helices in a cord is established by week 7 and retained throughout pregnancy (1). We hypothesize that developmental plasticity allows an intrinsic or stochastically occurring trigger to modify gene expression, enabling a change in helix formation and stabilization of the hUC arterial hypercoiling. Yet, the underlying mechanism and causality of gene expression on developmental origins of hypercoiling remain to be uncovered.

In conclusion, our investigation into the origin of hUC coiling shows that the helical structure of the hUC is primarily derived from the hUC arteries rather than the vein and Wharton's jelly. We demonstrated that the tunica media of the hUC arteries consist of two muscle cell layers with muscle fibers oriented in opposite directions. This unique arrangement is specific to the umbilical artery and is likely to enable coiling of the arteries. Showing that MZ

twin pairs can be highly discordant for hUC coiling, it is unlikely that the dominant factor for coiling is of genetic or parental origin. Comparing the cord arterial transcriptomic profile of these MZ twin pairs, we found 28 differentially expressed genes. This includes the up-regulation of genes associated with muscle fiber alignment, cell–cell interaction, regulation of the stromal collagen matrix, polarity, and axis formation that may enhance increased muscle fiber alignment in hUC arteries resulting in an increased number of helices. This gives a new twist on the existing explanations of the origin of hUC helices and provides a basis to further elucidate the biological origin of hUC coiling.

# Materials and Methods

### Ethical statement

hUCs from monozygotic (MZ) more specifically monochorionic twin pairs were collected at the Department of Obstetrics at the Leiden University Medical Center in the Netherlands with the ethical approval of the institutional medical ethical committee (P18.184). Written informed consent for the collection was obtained from all parents in the framework of the TwinLIFE study (International Clinical Trials Registry Platform ID NL7538) (68).

### Umbilical cord artery collection

After birth, the hUCs were cut from the chorionic plate of the chorioallantoic placenta (placenta) ~5–10 cm from the hUC insertion and transferred to a PBS solution supplemented with 0.38 µg/ml polymyxin B sulfate (Sigma-Aldrich), 20 µg/ml kanamycin (Gibco), 10 µg/ml penicillin/streptomycin (Lonza), and 1 µg/ml amphotericin B (Sigma-Aldrich). The hUCs were kept at 4°C for a maximum of 24 h until processing. The umbilical cord helices were retained after separation from the placenta, and the UCI was determined for all twin pairs included in the TwinLIFE study that were born between April 2021 and September 2021. The UCI was calculated by dividing the number of coils by the entire length of the retrieved hUC as described by reference 8. hUC arteries were collected when one of the hUCs displayed a UCI above 0.3 (*hypercoiled/case*) and the other hUC displayed a UCI below 0.3 (*normocoiled/control*). Discordant MZ twin pairs were therefore used as their own control group. To extract the hUC arteries, the collected hUC was cut into pieces of ~3 cm. All pieces were placed into a container with sterile PBS to wash off the remaining blood. Thereafter, one hUC piece was randomly retrieved from the container and cut longitudinally to reveal the hUC arteries within Wharton's jelly. Subsequently, the hUC arteries were extracted, again thoroughly rinsed with PBS to remove all the remaining blood, and partly used for histological assessment and partly snap-frozen in liquid nitrogen for DNA and RNA isolation.

### Hematoxylin/eosin (H&E) and elastin staining

For histological assessment, part of the hUCs and separately extracted hUC arteries were fixated in 4% formalin and embedded in paraffin blocks. Formalin-fixed, paraffin-embedded hUC tissues

were hematoxylin-and-eosin (H&E)– and Van Gieson's elastin–stained according to the standardized routine laboratory protocol. The hUCs and arteries were histologically evaluated as previously described by reference 69. These formalin-fixed, paraffin-embedded hUC arteries were subsequently used to cut 100 consecutive serial slides (thickness 4 μm), for digital reconstruction as described in the next section.

### Umbilical cord arterial digital reconstruction

Imaging of the consecutive sections was performed with the Pannoramic 250 Flash II slide scanner (3DHISTECH, Hungary) at a magnification of 40x. After acquisition, the images of the sections were aligned using custom MATLAB (version R2021b, MathWorks Inc.) scripts (available upon request), Libvips (70), and TrakEM2 (71), followed by histogram equalization (MATLAB).

### Isolation and culture of umbilical cord mesenchymal stromal cells

hUC-MSCs were isolated from the hUC using our robust and standardized method (72). In short, hUC pieces were placed on a petri dish with the inside of the cord facing down. Subsequently, culture medium consisting of minimum essential medium α (αMEM) GlutaMAX (Gibco) supplemented with 100 μg/ml penicillin/streptomycin (Gibco) and 5% PLTGOLD human platelet lysate (Merck) was added, and the dishes were incubated in a humidified atmosphere at 37°C with 5% $CO_2$. After hUC-MSC outgrowth, hUC-MSCs were dissociated from the petri dishes and expanded to passage 1. The culture medium was changed twice a week. Passage 1 in liquid nitrogen snap-frozen hUC-MSC pellets was used for RNA isolation.

### RNA/DNA isolation and quantification

DNA and RNA were extracted from the same randomly selected snap-frozen artery fragment using *Quick*-DNA/RNA Miniprep Plus Kit (Zymo Research), and RNA was isolated from the hUC-MSC pellets using *Quick*-DNA/RNA Microprep Plus Kit (Zymo Research) according to the manufacturer's protocol. DNA and RNA concentrations were assessed using Qubit dsDNA High Sensitivity Assay Kit and Qubit RNA Broad Range Assay Kit (Invitrogen), respectively (Invitrogen). The RNA integrity number (RIN) was determined for a subset of the samples using Agilent 2100 Bioanalyzer Instrument (Agilent RNA 6000 Nano Reagents) to ensure proficient RNA quality for RNA sequencing. To ensure proficient DNA quality and minimal degradation, a subset of the samples was placed on an agarose gel and assessed.

### RNA sequencing

Total RNA (50 μl of 25 ng/μl RNA in RNase/DNase-free water) was submitted to Macrogen Europe. After passing in-house quality control, RNA-sequencing libraries were prepared using the Illumina TruSeq Stranded mRNA library prep. Subsequently, barcoded libraries were sequenced with a depth of 40 million paired reads per sample and a read length of 150 bp on the NovaSeq 6000 (Illumina).

### RNA-sequencing analysis

RNA-sequencing files were processed using the opensource BIOWDL RNA-seq pipeline v5.1.0 (73) developed at the LUMC. This pipeline performs FASTQ preprocessing including quality control, quality trimming, and adapter clipping, RNA-seq read alignment, and read quantification. FastQC (v0.11.9) was used for checking raw read QC. Adapter clipping was performed using Cutadapt (v2.10) with default settings. RNA-Seq reads' alignment was performed using STAR (v2.7.5a) on the GRCh38 human reference genome. The gene read quantification was performed using HTSeq-count (v0.12.4) with the setting "–stranded = reverse." The gene annotation used for quantification was Ensembl version 105.

For the differential gene expression analysis, R v4.1.0 was used. The read count data of eight samples were labeled into two main groups (control: UCI < 0.3; hypercoiled: UCI >0.3), and each of the four MZ twin pairs was paired on familyID. During the preprocessing of raw count data, low-expressed genes were removed, by including only the transcripts with a $\log_2$CPM cutoff of 1 in at least 25% of the samples. The remaining counts (12,230 genes) were used as input to test for differential gene expression between *hypercoiled/case* and *normocoiled/control* samples using DESeq2 (v1.34.0). *Hypercoiled* and *normocoiled* samples were compared with familyID as a covariate to pair the comparison by family, an analysis equivalent to a paired *t* test. The Benjamini–Hochberg procedure was used to correct for multiple testing, and a false discovery rate (FDR) < 0.05 was considered statistically significant.

### Pathway enrichment analysis

The pathway enrichment analysis was based on four existing databases: GO Biological Process (2021) (74), BioPlanet (2019) (75), WikiPathways Human (2021) (76), and Hallmark Molecular Signature (2022) (77). The 28 differentially expressed genes ($P_{FDR}$ < 0.05) were used as input, and the 12,230 genes expressed in our dataset were used as background. All 28 differentially expressed genes and 10,785 of 12,230 background genes were annotated with at least one pathway in the four queried databases. Pathway enrichment P-values were adjusted for multiple testing using the Benjamini–Hochberg method, and only pathways with a $P_{FDR}$ < 0.05 were considered significantly enriched.

### DNA methylation

Total DNA (500 ng in 45 μl RNase/DNase-free water) was submitted to the Human Genotyping Facility of the Genetic Laboratory of the Department of Internal Medicine at the Erasmus Medical Center for DNA methylation measurement using the Illumina Infinium Methylation EPIC BeadChip array. Samples were run on the same array in an order randomized for coiling while keeping the twin pairs adjacent.

### DNA methylation analysis

The DNAmArray pipeline was used for preprocessing and quality control of the methylation data (78). Sample quality was assessed using visualizations, including MethylAid (79) plots, to detect

outlying or unreliable values. The data further underwent functional normalization, probe QC, and imputed missing values.

After quality control, the dataset contained DNA methylation at 852,836 CpGs and was annotated to the GRCh38/hg38 reference genome using InfiniumMethylation BeadChips Annotation Manifest of reference 80 (GENCODEv36, EPIC, https://zwdzwd.github.io/InfiniumAnnotation). Subsequently, CpGs located in the ENCODE blacklist regions (81) (8,838 CpGs), polymorphic probes according to reference 80 (95,453 CpGs) (Mask information, EPIC, https://zwdzwd.github.io/InfiniumAnnotation), and CpGs located on the X (18,703 CpGs), Y (99 CpGs), and M (7 CpGs) chromosomes were omitted leaving a dataset of 729,706 CpGs (80). To assess differential DNA methylation, a linear model was applied as follows: "CpGs ~ CoilingStatus + FamilyID" (CpG, beta-value at each CpG; CoilingStatus, binary variable Hypercoiled/Control; FamilyID, FamilyID of each MZ twin pair). CpGs were considered differentially methylated when the for-adjusted (FDR) *P*-values were <0.05. Statistical analysis, graphs, and figures were created using R Software version 4.1.3 and BioRender (https://biorender.com/).

## Data Availability

The raw data used for analysis in this study are available upon reasonable request from the corresponding author (LE van der Meeren). The data are not publicly available because of privacy restrictions.

## Supplementary Information

## Acknowledgements

We wish to thank all mothers and fathers who participated in the TwinLIFE study with their newborn twins. This work is funded by an Established Investigator grant from the Dutch Heart Foundation (2017T075 to BT Heijmans) and by a grant (Grant for Growth Innovation (GGI) to BT Heijmans) from Merck Healthcare KGaA, Darmstadt, Germany. The funders had no role in study design, data collection, analysis, decision to publish, or preparation of the article.

### Author Contributions

P Todtenhaupt: conceptualization, data curation, formal analysis, investigation, methodology, and writing—original draft, review, and editing.
TB Kuipers: formal analysis.
KL Dijkstra: data curation.
LM Voortman: formal analysis.
LA Franken: data curation.
JA Spekman: data curation.
TH Jonkman: formal analysis.
SG Groene: data curation.
AAW Roest: conceptualization and writing—review and editing.
MC Haak: conceptualization and writing—review and editing.
ET Verweij: data curation.
M van Pel: conceptualization and writing—review and editing.
E Lopriore: conceptualization and data curation.
BT Heijmans: conceptualization, funding acquisition, and writing—review and editing.
LE van der Meeren: conceptualization, data curation, and writing—original draft, review, and editing.

### Conflict of Interest Statement

The authors declare that they have no conflict of interest.

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
