## [Reviewer comments · Life Science Alliance]

Life Science Alliance

Twisting the theory on the origin of human umbilical cord coiling featuring monozygotic twins

Pia Todtenhaupt, Thomas B Kuipers, Kyra L Dijkstra, Lenard M Voortman, Laura A Franken, Jip A Spekman, Thomas H Jonkman, Sophie G Groene, Arno A W Roest, Monique C Haak, E Joanne T Verweij, Melissa van Pel, Enrico Lopriore, Bastiaan T Heijmans and Lotte Elisabeth van der Meeren

DOI: <https://doi.org/10.26508/lsa.202302543>

Corresponding author(s): Lotte Elisabeth van der Meeren (Leiden University Medical Center; Erasmus Medical Center; Leiden University Medical Center)

Review Timeline:

Submission Date:	2023-12-19
Editorial Decision:	2024-02-22
Revision Received:	2024-03-27
Editorial Decision:	2024-04-17
Revision Received:	2024-05-11
Editorial Decision:	2024-05-13
Revision Received:	2024-05-21
Accepted:	2024-05-22

Transaction Report:

February 22, 2024

Re: Life Science Alliance manuscript #LSA-2023-02543-T

Lotte E. van der Meeren
Leiden University Medical Center
Department of Pathology
Eindhovenweg 20
Leiden 2333 ZC Leiden
Netherlands

Dear Dr. van der Meeren,

Thank you for submitting your manuscript entitled "Twisting the theory on the origin of human umbilical cord coiling featuring monozygotic twins." to Life Science Alliance. The manuscript was assessed by expert reviewers, whose comments are appended to this letter. We invite you to submit a revised manuscript addressing the Reviewer comments.

Thank you for this interesting contribution to Life Science Alliance. We are looking forward to receiving your revised manuscript.

Sincerely,

B. MANUSCRIPT ORGANIZATION AND FORMATTING:

Reviewer #1 (Comments to the Authors (Required)):

Study Overview

The authors, Todtenhaupt et al., have presented a study which investigates hypercoiling of the human umbilical cord (hUC), the rationale being that hypercoiling is sometimes associated with fetal morbidity.

By comparing selected parameters on the umbilical cords of four pairs of monozygotic twins (N = 8 umbilical cords, N=16 arteries, N=8 veins), each pair of which exhibited "discordant" hUCs, i.e., one member of the pair exhibited normal coiling whilst the other member exhibited hypercoiling, variables such as genetics and environment were ostensibly controlled by referring to each pair's normocoiled hUC.

As a result, the authors claim insight into the origin of coiling within the arterial component of the hUC and, due to differences in gene expression between normo- and hypercoiled hUCs, offer similar insight into molecular pathways that might be involved in hypercoiling.

Summary of Major Concerns

As described in the next section, the manuscript, in its present state, lacks critical details regarding the cords used in this study: e.g., gross morphology of the cords and especially, methodology regarding coiling measurements and the specific pieces used. Missing also are the status of coiling along the length of the cords, sampling parameters relative to the (non-uniform) physiology of the umbilical cord along its length, and normocoiled control pairs. Also lacking is well-cited documentation regarding the frequencies of hypercoiling and their association with morbidity. Not only does this absence of methodological detail compromise the interpretation of the results, but without it, no other research group will be able to replicate this study. Taken together, I am sceptical of the conclusions drawn from this manuscript.

Specific Concerns

A study of this importance, especially in light of the precious material used, needs to be far better controlled consistent with all of the following parameters:

1. The coiling index and statistically evaluated distribution of coils along the length of the umbilical cord.

- How exactly was the umbilical coiling index (UCI) calculated? The authors state in the Methods (p. 17, lines 402-403) that, "The UCI was calculated by dividing the number of coils by the length of the retrieved hUC piece as described by Strong et al." What was the "retrieved piece"? Was it an unbiased selection, based on careful analysis and statistical evaluation of the distribution of the coils along the length of the cord, or was it any piece that appeared to have more coils than others?
- Is calculation of the UCI standardized across studies? If the standard method is to calculate the number of coils in just any retrieved piece without careful statistical analysis of the whereabouts of coils along the length, then the authors need to describe why this is acceptable rather than introduce a more reproducible and accurate method for measuring the UCI.
- Did the authors calculate the UCI before or after removal of the hUC from the chorion and, if so, is that standard?
- After separation from the chorion, did the hUC uncoil, or did the coils remain in place?
- What was the distribution of the coils along the cord? Were they clustered in any particularly physiologically important site (please see 3, below), or were they uniformly distributed?

2. Cord length. The authors do not discuss the length of the cords, and the possibility that the hypercoiled cord, if stretched out and uncoiled, was longer than the normocoiled one. Perhaps hypercoiling is due to excessive growth? Given that excessive

growth might have preceded hypercoiling, its molecular signature might not have been detected using the methods here.

- What was the length of each cord in each pair and how would that be measured, i.e., is the cord measured in situ, prior to separating it from the chorion and is the length of each coil taken into consideration?
- Was there any difference in the uncoiled length of the hUC in the discordant twins? If there was a difference in cord length, this needs to be addressed as part of the interpretation of results, and in terms of sampling site, below.

3. Sampling sites and physiology of the umbilical cord.

- Where along the length of the umbilical cords was sampling carried out? What were its dimensions and was the same piece used for all of the experiments? Or were different pieces used? The authors do not say; in the absence of this information, I can only assume that the authors used the same, possibly random piece for each hUC, thereby treating each cord's morphology and physiology as uniform - is it?

Variations in the cord's physiology along its length would undoubtedly be reflected in gene expression. It would not be surprising if the differences observed between normo- and hypercoiled hUC could be attributable to normal physiological differences at the sampling sites if efforts were not made to match those sampling sites along the length of the cord.

Sampling therefore should have been carried out at similar sites in each cord to account for the possibility of non-uniform physiology along its length, i.e., at some specific distance from its site of insertion into the chorion, at some specific distance from its site of insertion at the fetus, and/or perhaps at some fixed middle distance from either one of those sites.

- Sampling site gets back to the issue of the distribution of the coils and the length of the cord, above, i.e., how comparable were the samples along the length of the cord, especially if hypercoils are not evenly distributed, or the cord lengths are dissimilar? These potential differences further argue for sampling at a set distance e.g., from the chorion, and/or fetus and, ideally, at multiple sites.
- Given that the physiology of the cord is probably not uniform along its length (which again, needs to be addressed), and the sampling sites were not apparently controlled, how truly comparable are the molecular results within and between each set of twins?

4. Frequency of hypercoiling and morbidity. Due to lack of clear citations, the authors have not convinced this Reviewer that hypercoiling can be detrimental to fetal survival which was, presumably, the major rationale for undertaking this study.

- What is the normal frequency of hypercoiling in untwinned births within the demographic of the twins used in this study, and what is the frequency of morbidity in those pregnancies?
- What is the frequency of discordant cords in monozygotic twins? What is the frequency of concordant cords in monozygotic twins, both normo- and hypercoiled?
- What is the frequency with which hypercoiling is associated with fetal morbidity in discordant monozygotic twins? In concordant but hypercoiled twins?
- What was the outcome of the twins studied here? Were they normal? If normal, then results obtained via properly controlled sampling, as described above, would likely be meaningful. However, if the twin with hypercoiling ran into difficulties, then the molecular differences that the authors observed between normo- and hypercoiled hUCs would more likely be attributable to the detrimental effects of hypercoiling on the fetus and not necessarily to the normal biology of hypercoiling.

5. Normocoiled monozygotic twins. I understand the authors' rationale for using discordant twins, claiming that the normocoiled hUC can be used as the control. However, they nevertheless need to include pairs of monozygotic twins each member of which exhibits normocoiled hUCs of similar lengths with sampling carried out in a standard manner as proscribed above, for the following reason: the authors need to show whether the prediction that there would be no meaningful differences between normocoiled twins is correct - if there are molecular differences, then the results of the present study, which apparently also show such differences (though with all of the aforementioned caveats regarding the samples used), may not be significant.

6. Citations/attributions. Throughout the manuscript, statements are too often presented without citation, especially theories about hypercoiling. For example,

- Introduction. p.6, lines 121-123: What is the source (citation) of the conclusion that several hypotheses regarding the origin of helices were shelved and/or invalidated and why were these ideas discarded?
- Discussion. p. 11, lines 239-241 - What were the previous hypotheses about hypercoiling, why were they discarded, and in

what way do the authors' data refute/support them? p. 11, lines 247- 256 - The authors allude to the previous hypotheses, i.e., active or passive movement of the fetus, hemodynamics, etc., but the citation notes are poor and confusing - Please carefully document, with citations, the source of every statement made in this section. For example, line 252: "Similarly, a hemodynamic origin can be refuted." What is the source that has claimed the possibility of a hemodynamic origin and where can refutation be found?

7. Miscellany

Several more considerations:

- Materials and Methods p. 17, line 396: The authors state that the hUC was cut from the "placenta" - it is rather surprising that these authors do not know that the term "placenta" is shortened vernacular for the more formal "chorio-allantoic placenta", which is composed of BOTH the chorionic disk AND the umbilical cord. Thus, the authors cut the hUC from the chorionic disk - the chorionic disk alone is not the "placenta". Please use correct terminology.
- Results p. 8, line 175, section entitled "Hypercoiled umbilical arteries display a distinct transcriptomic profile": Shouldn't the authors have analysed 16 arteries, rather than 8, given that there are two arteries per cord? How did they distinguish members of each pair within a single hUC, or was each pair considered to be physiologically identical and what was the basis for that supposition?

Reviewer #2 (Comments to the Authors (Required)):

Excellent study utilizing advanced techniques. Major contribution in a field that stagnated for a long time.

Reviewer #3 (Comments to the Authors (Required)):

The manuscript uses a combination of tissue dissection, histological analysis and genome sequencing to explore the origin of the umbilical cord coiling in the human pregnancy. In particular, the authors observed a divergence in the umbilical cord coiling between otherwise identical monozygotic twins. The authors propose that unique mechanical features of the umbilical arterial (UA) wall are the primary origin of cord coiling.

Overall, this is an intriguing, highly original, timely and well-written study that addresses a long-standing open question in reproductive vascular physiology, and which has the potential for significant clinical impact. I congratulate the authors on their work; however, despite these strengths, the manuscript would benefit from resolving a few clarification and presentation issues.

"The umbilical arteries consist of two muscle fiber layers, the outer and the inner."

Please note that Gasser and co-workers (J. R. Soc. Interface 2005, doi:10.1098/rsif.2005.0073) have also observed that in many systemic arteries collagen fibres are arranged into two helically distributed families in the medial portion of the wall, and a more random orientation in the adventitial and intimal layers. Could you comment on the relevance of this for the umbilical arteries?

To further support your angio-genetic / bio-mechanical argument on the origin of the UA coiling, I wonder if you could correlate the overall handedness of the cord coiling (clockwise vs. anticlockwise) and the relative orientation of the UA medial muscle layers (e.g., clockwise inner and anticlockwise outer)?

"Remarkably, the helices were retained in each of the two arteries upon removal while ... the vein of the hUC appeared to have lost the distinctive coiling characteristic after removal of the arteries..."

What about the loss of pressure in the umbilical vein as another major possible contributing factor? For instance, what is known about the coiling of the cords with fetal hyper/hypotension over gestation. Have you clamped the umbilical cord ends immediately before or after delivery?

Specific points:

Abstract:"Surprisingly, we observed that genetically identical twins can be discordant ..., excluding genetic, environmental, and parental origins of hUC coiling."

I suggest avoiding an undue focus on the "non-environmental" origins of coiling, since it contradicts a later statement (page 15, line 354) that this study could be underpowered to discriminate the impact of epigenetic as well as mechanical differences between the twins on the UA.

Fig. 2: To what extent could the collapse (and a possible contraction) of the UA vascular wall influence the perceived partitioning of the medial muscular layer into two?

* Suppl. Table 1: Could you please add further key details on the pregnancy outcomes, such as maternal age & parity, Caesarian or vaginal delivery; birth weight and/or birth weight centile?

* Page 39: move lines 849-852 to the Table 1 caption at the top of the page.

Ref#: LSA-2023-02543-T

Manuscript – Twisting the theory on the origin of human umbilical cord coiling featuring monozygotic twins.

Point-by-point response

Reviewer #1:

Study Overview

The authors, Todtenhaupt et al., have presented a study which investigates hypercoiling of the human umbilical cord (hUC), the rationale being that hypercoiling is sometimes associated with fetal morbidity.

By comparing selected parameters on the umbilical cords of four pairs of monozygotic twins (N = 8 umbilical cords, N=16 arteries, N=8 veins), each pair of which exhibited "discordant" hUCs, i.e., one member of the pair exhibited normal coiling whilst the other member exhibited hypercoiling, variables such as genetics and environment were ostensibly controlled by referring to each pair's normocoiled hUC.

As a result, the authors claim insight into the origin of coiling within the arterial component of the hUC and, due to differences in gene expression between normo- and hypercoiled hUCs, offer similar insight into molecular pathways that might be involved in hypercoiling.

Summary of Major Concerns

As described in the next section, the manuscript, in its present state, lacks critical details regarding the cords used in this study: e.g., gross morphology of the cords and especially, methodology regarding coiling measurements and the specific pieces used. Missing also are the status of coiling along the length of the cords, sampling parameters relative to the (non-uniform) physiology of the umbilical cord along its length, and normocoiled control pairs. Also lacking is well-cited documentation regarding the frequencies of hypercoiling and their association with morbidity. Not only does this absence of methodological detail compromise the interpretation of the results, but without it, no other research group will be able to replicate this study. Taken together, I am sceptical of the conclusions drawn from this manuscript.

We thank the reviewer for the comments and made substantial changes to improve the manuscript based on the reviewers' recommendation. Below, we provide a detailed response to the reviewer's comments, outlining how we addressed each suggestion and concern raised. This includes addressing the major concerns mentioned above.

Manuscript – Twisting the theory on the origin of human umbilical cord coiling featuring monozygotic twins.

Specific Concerns

A study of this importance, especially in light of the precious material used, needs to be far better controlled consistent with all of the following parameters:

1. The coiling index and statistically evaluated distribution of coils along the length of the umbilical cord.

- How exactly was the umbilical coiling index (UCI) calculated? The authors state in the Methods (p. 17, lines 402-403) that, "The UCI was calculated by dividing the number of coils by the length of the retrieved hUC piece as described by Strong et al." What was the "retrieved piece"? Was it an unbiased selection, based on careful analysis and statistical evaluation of the distribution of the coils along the length of the cord, or was it any piece that appeared to have more coils than others?

We thank the reviewer for highlighting the need to improve the clarity on this matter. To do so, we added an explanation to the methods section in which we define that the 'retrieved hUC piece' is the entire cord umbilical cord cut from the placenta leaving ~ 5-10 cm at the placenta. Therefore, we can confirm that the coiling index was calculated unbiased and without selection for a specific piece of the umbilical cord.

[Line 398-399: *After birth, the hUCs were cut from the chorio-allantoic placenta (placenta) approximately 5-10 cm from the hUC insertion and...;*

Line 406-407: *The UCI was calculated by dividing the number of coils by the entire length of the retrieved hUC ...]*

- Is calculation of the UCI standardized across studies? If the standard method is to calculate the number of coils in just any retrieved piece without careful statistical analysis of the whereabouts of coils along the length, then the authors need to describe why this is acceptable rather than introduce a more reproducible and accurate method for measuring the UCI.

The calculation of the UCI is indeed standardized across studies. It is standard practice to calculate the umbilical cord index based on the entire length of the umbilical cord. This is substantiated by the following references:

- (A) Chan, J. S., & Baergen, R. N. (2012). Gross umbilical cord complications are associated with placental lesions of circulatory stasis and fetal hypoxia. *Pediatric and Developmental Pathology*, 15(6), 487-494.
- (B) Benirschke K, Burton GJ, Baergen RN. *Pathology of the Human Placenta*. 6th ed. Berlin, Germany: Springer, 2012
- (C) Dutman, A. C., & Nikkels, P. G. (2015). Umbilical hypercoiling in 2nd-and 3rd-trimester intrauterine fetal death. *Pediatric and Developmental Pathology*, 18(1), 10-16.

Manuscript – Twisting the theory on the origin of human umbilical cord coiling featuring monozygotic twins.

- Did the authors calculate the UCI before or after removal of the hUC from the chorion and, if so, is that standard?

We calculate the umbilical cord index as a standard clinical procedure. We did so after removal from the placenta on the intact umbilical cord before any additional experiments. After removing the arteries from the Wharton's Jelly, the tissue was further processed. As the number helices of the umbilical cord do not change after removal from the placenta, the umbilical cord index can be assessed before and after separation.

- After separation from the chorion, did the hUC uncoil, or did the coils remain in place?

After separation from the placenta, the umbilical cord maintains its helical structure. To clarify this point, we have included this information in the methods section.

[Line 403-404: *The umbilical cord helices were retained after separation from the placenta and the umbilical cord index (UCI) was determined for all twin pairs ...*]

- What was the distribution of the coils along the cord? Were they clustered in any particularly physiologically important site (please see 3, below), or were they uniformly distributed?

The helical patterns were quite evenly spread, forming what is commonly referred to as a 'rope pattern'. To be transparent about the umbilical cords we used, the manuscript includes images of all the umbilical cords prior to processing. Additionally, we have added this information to the result section of the manuscript.

[Page 34: *Figure 1: Overview of the included umbilical cords*]

[Line 144-146: *The length of the hUC was comparable between the individuals of each of the four twin pairs and was within normal limits (30-70 cm)ⁱ. The helical pattern was evenly spread throughout the hUC.*]

ⁱBenirschke K, Burton GJ, Baergen RN. Pathology of the Human Placenta. 6th ed. Berlin, Germany: Springer, 2012

2. Cord length. The authors do not discuss the length of the cords, and the possibility that the hypercoiled cord, if stretched out and uncoiled, was longer than the normocoiled one. Perhaps hypercoiling is due to excessive growth? Given that excessive growth might have preceded hypercoiling, its molecular signature might not have been detected using the methods here.

We appreciate the reviewer's insightful question and theory. We do not assume that hypercoiling is due to excessive growth, given Extensively Long Umbilical Cords (ELUCs) were associated with a variety of placental pathologies such as true knots but not with hypercoiling in a study conducted by Baergen et al.

Baergen, R. N., Malicki, D., Behling, C., & Benirschke, K. (2001). Morbidity, mortality, and placental pathology in excessively long umbilical cords: retrospective study. *Pediatric and Developmental Pathology*, 4(2), 144-153.

Manuscript – Twisting the theory on the origin of human umbilical cord coiling featuring monozygotic twins.

Additionally, our attention was directed towards examining the presence and molecular characteristics of hypercoiling. If hypercoiling were driven by excessive growth, we would expect finding differential expression of genes associated with growth or proliferation. Yet, such differential expression was not observed.

- What was the length of each cord in each pair and how would that be measured, i.e., is the cord measured in situ, prior to separating it from the chorion and is the length of each coil taken into consideration?

The umbilical cord index was measured after separation from the placenta. As the umbilical cord index is calculated by dividing the number of coils by the length of the umbilical cord, the length of each coil is taken into consideration. To address this in the manuscript, we have included both the number of coils and the length of the umbilical cords in Supplemental Table 1.

[Supplemental Information page 6:

Supplemental Table 1: Donor and umbilical cord characteristics of monozygotic twin pairs discordant for coiling]

- Was there any difference in the uncoiled length of the hUC in the discordant twins? If there was a difference in cord length, this needs to be addressed as part of the interpretation of results, and in terms of sampling site, below.

The length of the umbilical cord between the discordant twins was comparable.

[Line 144-146: The length of the hUC was comparable between the individuals of each of the four twin pairs and was within normal limits (30-70 cm)ⁱ.]

ⁱBenirschke K, Burton GJ, Baergen RN. Pathology of the Human Placenta. 6th ed. Berlin, Germany: Springer, 2012

3. Sampling sites and physiology of the umbilical cord.

- Where along the length of the umbilical cords was sampling carried out? What were its dimensions and was the same piece used for all of the experiments? Or were different pieces used? The authors do not say; in the absence of this information, I can only assume that the authors used the same, possibly random piece for each hUC, thereby treating each cord's morphology and physiology as uniform - is it?

We thank the reviewer for bringing attention to the requirement for further clarification. The reviewer is indeed correct, we used the same randomly selected piece of artery for the simultaneous isolation of DNA and RNA for all molecular measurements. The helical pattern was evenly spread throughout the umbilical cord, forming what is commonly referred to as a 'rope pattern'. Therefore, the sampling site is unbiased, and we can optimally link the methylomic and transcriptomic profiles. To improve clarity on this matter, we made some changes in the manuscript.

Manuscript – Twisting the theory on the origin of human umbilical cord coiling featuring monozygotic twins.

[Line 445-446: *DNA and RNA were extracted from the same randomly selected snap-frozen artery fragment ...*]

Variations in the cord's physiology along its length would undoubtedly be reflected in gene expression. It would not be surprising if the differences observed between normo- and hypercoiled hUC could be attributable to normal physiological differences at the sampling sites if efforts were not made to match those sampling sites along the length of the cord.

Sampling therefore should have been carried out at similar sites in each cord to account for the possibility of non-uniform physiology along its length, i.e., at some specific distance from its site of insertion into the chorion, at some specific distance from its site of insertion at the fetus, and/or perhaps at some fixed middle distance from either one of those sites.

We appreciate this interesting thought raised by the reviewer. To our knowledge, the potential alteration in the molecular profile of the umbilical cord or umbilical cord arteries with increasing distance from the placenta has not been investigated. Since the selection of the umbilical cord segment for artery extraction was random, not within the first 5-10 cm of the connection to the placenta and coiling was uniformly distributed along the cord's length, we expect no interference with our comparative analysis.

- Sampling site gets back to the issue of the distribution of the coils and the length of the cord, above, i.e., how comparable were the samples along the length of the cord, especially if hypercoils are not evenly distributed, or the cord lengths are dissimilar? These potential differences further argue for sampling at a set distance e.g., from the chorion, and/or fetus and, ideally, at multiple sites.

Considering the uniform distribution of helices throughout the length of the umbilical cord as a rope pattern we also expect the samples along the cord to be highly comparative.

[see Page 34: Figure 1: Overview of the included umbilical cords].

- Given that the physiology of the cord is probably not uniform along its length (which again, needs to be addressed), and the sampling sites were not apparently controlled, how truly comparable are the molecular results within and between each set of twins?

The physiology of the cord is rather uniform along its length [see Page 34: Figure 1: Overview of the included umbilical cords]. To highlight this, we added this information to the result section of the manuscript.

[Line 144-146: *The length of the hUC was comparable between the individuals of each of the four twin pairs and was within normal limits (30-70 cm)ⁱ. The helical pattern was evenly spread throughout the hUC.*]

ⁱBenirschke K, Burton GJ, Baergen RN. Pathology of the Human Placenta. 6th ed. Berlin, Germany: Springer, 2012

Because of the random selection of the hUC artery segments and the even distribution of helices along the hUC, we do not expect the sampling site to affect the intra-pair comparison of the molecular profiles.

Manuscript – Twisting the theory on the origin of human umbilical cord coiling featuring monozygotic twins.

4. Frequency of hypercoiling and morbidity. Due to lack of clear citations, the authors have not convinced this Reviewer that hypercoiling can be detrimental to fetal survival which was, presumably, the major rationale for undertaking this study.

We thank the reviewer for highlighting this aspect. While the primary aim of our study was to investigate the origin of umbilical cord helices, it is crucial to acknowledge the broader implications of our findings in the context of clinical outcomes. Understanding umbilical cord helices is of significant interest due to their association with adverse clinical outcomes. These outcomes encompass various morbidities, such as fetal growth restriction and fetal distress, and in extreme cases fetal demise. This association is supported by evidence of several conducted studies, referred to in the introductory section of our manuscript. To make sure that these points are clearly stated and supported by explicit citations in the manuscript, we added additional citations.

[Line 113-116: *Numerous studies linked an abnormal hUC coiling intensity to pre and perinatal morbidity and mortality*^{2,6,7,10,11}. *Hypercoiling is associated with fetal growth restriction and fetal distress resulting in (planned) premature delivery, a decreased Apgar score at 5 minutes or even fetal demise*^{6,7}]

- ² Hayes DJL, Warland J, Parast MM, et al. Umbilical cord characteristics and their association with adverse pregnancy outcomes: A systematic review and meta-analysis. Ryckman KK, ed. PLoS One. 2020;15(9):e0239630. doi:10.1371/journal.pone.0239630
- ⁶ Pergialiotis V, Kotrogianni P, Koutaki D, Christopoulos-Timogiannakis E, Papantoniou N, Daskalakis G. Umbilical cord coiling index for the prediction of adverse pregnancy outcomes: a meta-analysis and sequential analysis. J Matern Neonatal Med. 2020;33(23):4022-4029. doi:10.1080/14767058.2019.1594187
- ⁷ Subashini G, Anitha C, Gopinath G, Ramyathangam K. A Longitudinal Analytical Study on Umbilical Cord Coiling Index as a Predictor of Pregnancy Outcome. Cureus. March 2023. doi:10.7759/cureus.35680
- ¹⁰ Machin GA, Ackerman J, Gilbert-Barnes E. Abnormal Umbilical Cord Coiling is Associated with Adverse Perinatal Outcomes. Pediatr Dev Pathol. 2000;3(5):462-471. doi:10.1007/s100240010103
- ¹¹ Kashanian M, Akbarian A, Kouhpayehzadeh J. The umbilical coiling index and adverse perinatal outcome. Int J Gynecol Obstet. 2006;95(1):8-13. doi:10.1016/j.ijgo.2006.05.029

• What is the normal frequency of hypercoiling in untwinned births within the demographic of the twins used in this study, and what is the frequency of morbidity in those pregnancies?

We thank the reviewer for his inquiry. As an illustration, in a study examining un-twinned births, hypercoiling was observed in 12.4% of the 699 cases (Kashanian, M., A. Akbarian, and J. Kouhpayehzadeh. "The umbilical coiling index and adverse perinatal outcome." International Journal of Gynecology & Obstetrics 95.1 (2006): 8-13.). However, it is important to note that the frequency of hypercoiling can vary among different studies. Similarly, the frequency of morbidity varies depending on the study and specific type of morbidity including intrauterine growth restriction, decreased APGAR score, or fetal demise.

• What is the frequency of discordant cords in monozygotic twins? What is the frequency of concordant cords in monozygotic twins, both normo- and hypercoiled?

Studies investigating the frequency of umbilical cord coiling have predominantly focused on singleton pregnancies. To our knowledge, large epidemiological studies specifically examining uncomplicated monozygotic twins have not been conducted. Existing research suggests that umbilical cords in twin pairs tend to be slightly shorter and more coiled compared to singleton pregnancies (Edmonds HW. The spiral twist of the normal umbilical cord in twins and in singletons. Am J Obstet Gynecol 1954;67:102–120). However, it is important to note that twin pairs are often viewed as individual entities rather than directly comparing cord characteristics within the same pair.

Manuscript – Twisting the theory on the origin of human umbilical cord coiling featuring monozygotic twins.

- What is the frequency with which hypercoiling is associated with fetal morbidity in discordant monozygotic twins? In concordant but hypercoiled twins?

To our knowledge, there is a lack of large-scale epidemiological studies specifically comparing fetal morbidity in monozygotic twins in relation to the umbilical cord index.

- What was the outcome of the twins studied here? Were they normal? If normal, then results obtained via properly controlled sampling, as described above, would likely be meaningful. However, if the twin with hypercoiling ran into difficulties, then the molecular differences that the authors observed between normo- and hypercoiled hUCs would more likely be attributable to the detrimental effects of hypercoiling on the fetus and not necessarily to the normal biology of hypercoiling.

We appreciate the reviewer's question. We have thoroughly examined the pre-, peri-, and post-natal health of the twins and did not observe significant disparities in morbidities or adverse effects of hypercoiling. As an illustration, the APGAR score, which measures the overall health and vitality of newborns based on several criteria, showed no differences between the twins nine minutes after birth. To address the reviewers comment, we have included in the results section that we did not find increased morbidity in the twin with the hypercoiled umbilical cord.

[Line 146-148: *We did not observe an increased morbidity in the twin with the hypercoiled umbilical cord compared to its co-twin.*]

5. Normocoiled monozygotic twins. I understand the authors' rationale for using discordant twins, claiming that the normocoiled hUC can be used as the control. However, they nevertheless need to include pairs of monozygotic twins each member of which exhibits normocoiled hUCs of similar lengths with sampling carried out in a standard manner as proscribed above, for the following reason: the authors need to show whether the prediction that there would be no meaningful differences between normocoiled twins is correct - if there are molecular differences, then the results of the present study, which apparently also show such differences (though with all of the aforementioned caveats regarding the samples used), may not be significant.

We appreciate the reviewer's thoughtful consideration of our study. Our primary objective was to explore the origin of umbilical cord helices. By comparing hypercoiled umbilical cord arteries with normo-coiled arteries, we aimed to identify genes implicated in hypercoiling. Our findings revealed a subset of genes differentially expressed, which are involved in vascular development, cell-cell interaction, polarity, and axis formation, potentially contributing to the increased number of observed helices. We anticipate that umbilical cords with similar coiling indices would exhibit fewer transcriptomic differences, particularly for these genes. While we understand the suggestion to include pairs of monozygotic twins with identical coiling indices and lengths, it poses challenges in feasibility. It would require multiple pairs meeting such strict criteria, which may be impractical to obtain. Additionally, such an approach would essentially serve as a validation step for our findings. It would require a considerable allocation of resources without significantly enhancing our understanding. Therefore, we decided to not pursue the suggestion at this time.

Manuscript – Twisting the theory on the origin of human umbilical cord coiling featuring monozygotic twins.

6. Citations/attributions. Throughout the manuscript, statements are too often presented without citation, especially theories about hypercoiling. For example,

- Introduction. p.6, lines 121-123: What is the source (citation) of the conclusion that several hypotheses regarding the origin of helices were shelved and/or invalidated and why were these ideas discarded?

We thank the reviewer for highlighting the need for including additional citations regarding the theory of hypercoiling. We have updated our citations throughout the manuscript, specifically focusing on the theories of hypercoiling. It is an observation of the authors that none of the current hypotheses were thus far validated. To support this conclusion, we have included relevant literature stating the same.

- Discussion. p. 11, lines 239-241 - What were the previous hypotheses about hypercoiling, why were they discarded, and in what way do the authors' data refute/support them? p. 11, lines 247- 256 - The authors allude to the previous hypotheses, i.e., active or passive movement of the fetus, hemodynamics, etc., but the citation notes are poor and confusing - Please carefully document, with citations, the source of every statement made in this section. For example, line 252: "Similarly, a hemodynamic origin can be refuted." What is the source that has claimed the possibility of a hemodynamic origin and where can refutation be found?

To address this comment, we have meticulously included citations to source all statements in the relevant section.

7. Miscellany

Several more considerations:

- Materials and Methods p. 17, line 396: The authors state that the hUC was cut from the "placenta" - it is rather surprising that these authors do not know that the term "placenta" is shortened vernacular for the more formal "chorio-allantoic placenta", which is composed of BOTH the chorionic disk AND the umbilical cord. Thus, the authors cut the hUC from the chorionic disk - the chorionic disk alone is not the "placenta". Please use correct terminology.

As the reviewer accurately pointed out, "chorio-allantoic placenta" is indeed the original developmental term for the mammalian placenta. While this formal terminology is technically correct, it is uncommonly used in practice. The shortened term "placenta" is much more prevalent in literature and books by renowned perinatal pathologists, such as the comprehensive work on placental pathology (Benirschke K, Burton GJ, Baergen RN. Pathology of the Human Placenta). To ensure clarity for a wide audience, we have opted to use the prevailing shortened vernacular "placenta". In response to the reviewer's valid observation, we have also added clarification in the methods section of the manuscript, explicitly stating that "placenta" refers to the "chorio-allantoic placenta".

[Line 398-399: *After birth, the hUCs were cut from the chorio-allantoic placenta (placenta) approximately 5-10 cm from the hUC insertion and...*]

Manuscript – Twisting the theory on the origin of human umbilical cord coiling featuring monozygotic twins.

- Results p. 8, line 175, section entitled "Hypercoiled umbilical arteries display a distinct transcriptomic profile": Shouldn't the authors have analysed 16 arteries, rather than 8, given that there are two arteries per cord? How did they distinguish members of each pair within a single hUC, or was each pair considered to be physiologically identical and what was the basis for that supposition?

We appreciate the reviewer for bringing up this valid point. Indeed, each umbilical cord contains two umbilical cord arteries. Our decision to analyze only one artery per umbilical cord was based on the similarities between the two arteries. Both arteries typically exhibit the retained helical pattern. In our study, we aimed to unravel the mechanism underlying umbilical cord artery coiling. We expect the mechanism for coiling to be the same in both arteries and considered it sufficient to analyze one of the two arteries in this comparative investigation. However, we acknowledge that comparing both arteries may be relevant for future research questions, and we will consider this aspect in subsequent studies.

Manuscript – Twisting the theory on the origin of human umbilical cord coiling featuring monozygotic twins.

Reviewer #2:

Excellent study utilizing advanced techniques. Major contribution in a field that stagnated for a long time.

We highly appreciate the compliments of the reviewer and are pleased to hear that our work is viewed as a significant contribution. It is encouraging to see that the reviewer also recognizes that after all these years of stagnation in the field, our research marks the beginning of a new phase of investigation that hopefully will inspire other colleagues to continue with research on the placenta and umbilical cord.

Manuscript – Twisting the theory on the origin of human umbilical cord coiling featuring monozygotic twins.

Reviewer #3:

The manuscript uses a combination of tissue dissection, histological analysis and genome sequencing to explore the origin of the umbilical cord coiling in the human pregnancy. In particular, the authors observed a divergence in the umbilical cord coiling between otherwise identical monozygotic twins. The authors propose that unique mechanical features of the umbilical arterial (UA) wall are the primary origin of cord coiling.

Overall, this is an intriguing, highly original, timely and well-written study that addresses a long-standing open question in reproductive vascular physiology, and which has the potential for significant clinical impact. I congratulate the authors on their work; however, despite these strengths, the manuscript would benefit from resolving a few clarification and presentation issues.

We thank the reviewer for the due diligence and valuable review of our manuscript, including an acknowledgement of its strengths. We are pleased to hear that our study is seen as intriguing, highly original, and potentially impactful. In light of the reviewer's feedback, we have addressed the identified clarification and presentation issues. Below, we present our detailed response to the specific suggestions made.

"The umbilical arteries consist of two muscle fiber layers, the outer and the inner."
Please note that Gasser and co-workers (J. R. Soc. Interface 2005, doi:10.1098/rsif.2005.0073) have also observed that in many systemic arteries collagen fibres are arranged into two helically distributed families in the medial portion of the wall, and a more random orientation in the adventitial and intimal layers. Could you comment on the relevance of this for the umbilical arteries?

We thank the reviewer for drawing attention to this publication. Gasser et al. evaluated the kinematics and elasticity of the arteries. They also acknowledge the presence of helically distributed fibers within the muscle layers. This underscores our hypothesis of the presence of crossing alignment of two muscle layers due to alignment of the fibers. We have incorporated the reference in our manuscript.

[Line 320-321: *Luis Martinez-Lemus and Gasser et al. extensively described the normal appearance of the arterial tunica media in the human body.*;

Line 325-326: *The collagen fibers in the tunica media are arranged in a helical structure.*]

To further support your angio-genetic / bio-mechanical argument on the origin of the UA coiling, I wonder if you could correlate the overall handedness of the cord coiling (clockwise vs. anticlockwise) and the relative orientation of the UA medial muscle layers (e.g., clockwise inner and anticlockwise outer)?

We appreciate this interesting comment raised by the reviewer. It was observed by Fletcher and Lacro (Am J Obstet Gynecol. 1987), that the counterclockwise (left) spiral typically exceeds the clockwise (right) spiral, a trend also reflected in our study population of eight individuals, where seven exhibited a left twist. The lack of association between the direction of helices and adverse events led us to dismiss this aspect in our study. However, the notion of potential differences in orientation between the inner and outer layers is thought-provoking. A study design specifically focusing on selecting samples with both left and right twists may be better suited to explore this intriguing suggestion further.

Manuscript – Twisting the theory on the origin of human umbilical cord coiling featuring monozygotic twins.

"Remarkably, the helices were retained in each of the two arteries upon removal while ... the vein of the hUC appeared to have lost the distinctive coiling characteristic after removal of the arteries..."

What about the loss of pressure in the umbilical vein as another major possible contributing factor? For instance, what is known about the coiling of the cords with fetal hyper/hypotension over gestation. Have you clamped the umbilical cord ends immediately before or after delivery?

We thank the reviewer for highlighting this important point. There umbilical cords are clamped after birth. Although loss of pressure in the umbilical vein is an important variable to consider, the umbilical vein appears never coiled. At ultrasound, during cross evaluation at the department of pathology after delivery and formalin fixation, the arteries are helices around a straight vein. We agree with the reviewer that this is not clearly stated in the manuscript, and we have made according changes.

[Line 151-154: *Remarkably, the helices were retained in each of the two arteries upon removal. The remaining hUC containing Wharton's Jelly and the umbilical vein exhibit no inherent coiling, suggesting that the source of umbilical cord coiling originates solely from the umbilical cord arteries.*]

Specific points:

Abstract: "Surprisingly, we observed that genetically identical twins can be discordant ..., excluding genetic, environmental, and parental origins of hUC coiling."

I suggest avoiding an undue focus on the "non-environmental" origins of coiling, since it contradicts a later statement (page 15, line 354) that this study could be underpowered to discriminate the impact of epigenetic as well as mechanical differences between the twins on the UA.

We agree with the reviewer that we can indeed not exclude all environmental factors to underlie umbilical cord coiling. Yet, as we observed MC twin pairs discordant for umbilical cord coiling many environmental origins can be dismissed considering a large part of their 'environment' is shared including the womb and parental factors. 'Environment' can compose of many different aspects that may also be discordant between genetically identical twin pairs which is why we nuanced this throughout the manuscript (including the abstract) and mention that many/majority of environmental origins can be dismissed as the origins of coiling. Thereby we avoid an undue focus on 'non-environmental' origins of umbilical cord coiling.

e.g. [Line 73-75: *Surprisingly, we observed that genetically identical twins can be discordant for hUC coiling, excluding genetic, many environmental, and parental origins of hUC coiling.*]

Fig. 2: To what extent could the collapse (and a possible contraction) of the UA vascular wall influence the perceived partitioning of the medial muscular layer into two?

We generally do not expect that the collapse of the vascular wall has an effect on the perceived partitioning of the two muscle cell layers in the tunica media. The observed layers are anatomical differences rather than a physiological attribute introduced by the collapse or contraction of the umbilical artery. Additionally, if the collapse or contraction of the umbilical

Manuscript – Twisting the theory on the origin of human umbilical cord coiling featuring monozygotic twins.

artery influenced the partitioning of the layers, we would expect this to affect hypercoiled and normo-coiled umbilical arteries to the same extent.

* Suppl. Table 1: Could you please add further key details on the pregnancy outcomes, such as maternal age & parity, Caesarian or vaginal delivery; birth weight and/or birth weight centile?

As requested, we have added key details on the pregnancy outcomes to supplemental table 1 including maternal age, parity, Caesarian/Vaginal delivery, and birthweight.

[Supplemental Information page 6:

Supplemental Table 1: Donor and umbilical cord characteristics of monozygotic twin pairs discordant for coiling]

* Page 39: move lines 849-852 to the Table 1 caption at the top of the page.

As requested, we have moved this section to the Table 1 caption at the top of the page.

April 17, 2024

Re: Life Science Alliance manuscript #LSA-2023-02543-TR

Dr. Lotte E. van der Meeren
Leiden University Medical Center
Department of Pathology
Eindhovenweg 20
Leiden 2333 ZC Leiden
Netherlands

Dear Dr. van der Meeren,

Thank you for submitting your revised manuscript entitled "Twisting the theory on the origin of human umbilical cord coiling featuring monozygotic twins." to Life Science Alliance. The manuscript has been seen by the original reviewers whose comments are appended below, and some important issues remain.

We are open to one additional short round of revision. Please note that I will expect to make a final decision without additional reviewer input upon re-submission.

Please submit the final revision within one month, along with a letter that includes a point by point response to the remaining reviewer comments.

To upload the revised version of your manuscript, please log in to your account: <https://lsa.msubmit.net/cgi-bin/main.plex>
You will be guided to complete the submission of your revised manuscript and to fill in all necessary information.

B. MANUSCRIPT ORGANIZATION AND FORMATTING:

Sincerely,

Reviewer #1 (Comments to the Authors (Required)):

Reviewer Summary of the Authors' Rebuttal:

In this study, the authors conclude that some unknown property of the arteries is responsible for umbilical coiling. This conclusion is based on their having "randomly" sampled and compared the arterial vessels found within each of four sets of discordant twins, i.e., where one member of the pair exhibits normo-coiling and the other hyper-coiling, which is sometimes associated with fetal morbidity (though not here, as the authors have now mentioned).

The most important figure in this manuscript is Figure 1, which shows the umbilical cords of the four pairs of discordant twins. It forms the cornerstone of this study. What continues to confuse and has not yet been satisfactorily explained is how the authors "randomly" sampled the cords. Whilst the morphology of all four of the abnormal hyper-coiled cords looks generally uniform along their length, that of the normo-coiled cords is highly variable. The authors point out the former as their basis for "random" sampling but ignore the latter. This begs the question: What was sampled in these cords?

To highlight the importance of this omission, each of the distinct morphologies in the normo-coiled cords could have produced a different result when compared against the hypercoils. Thus, were the coils of the normo-coiled cords used? Or the non-coiled parts? If one morphological type was selected over the other, this would hardly constitute "random" sampling. Each morphology could have produced a different result; therefore, each type of morphology should have been tested independently. Then, if all the comparisons were identical, "random" sampling, presumably anywhere along the length of the cord (but why do I have to guess?), might be appropriate between the normo- and hyper-coiled cords.

The authors' conclusions raise another issue that they did not address: if the arteries are responsible for creating umbilical coils, why are there so many fewer coils in the normo-coiled cords? After all, both cords have arteries running through them. The authors ignored the possibility of physiological variation in arterial function along the length of the cords, treating the cord as a uniformly inert pipe without any scientific justification. (Physiological variability would further argue against use of "random" sampling as the best approach for this study.)

Two more points. Firstly, the authors brush aside this Reviewer's suggestion that normal controls, i.e., concordant monozygotic twins, be included, claiming "impracticality". However difficult to achieve, they could nonetheless have addressed the strengths - or weaknesses - of this potentially critical control in the revised version. Secondly, it is further exasperating that the authors justify their incorrect use of the term "placenta" instead of "chorion" by citing usage of the former by Kurt Benirschke. Given what we have learned about the developmental biology of the placenta over the past several decades, that terminology is outdated and incorrect.

In summary, not only does the authors' lack of a detailed protocol regarding "random" sampling mean that their experiments can never be replicated by others, but their obfuscation means that readers cannot properly evaluate their data, leaving us to take them at their word. This is unacceptable.

Specific comments to the authors' rebuttal can be found below.

Authors' Rebuttal and Reviewer's Comments:

1. The coiling index and statistically evaluated distribution of coils along the length of the umbilical cord.

- How exactly was the umbilical coiling index (UCI) calculated? The authors state in the Methods (p. 17, lines 402-403) that, "The UCI was calculated by dividing the number of coils by the length of the retrieved hUC piece as described by Strong et al." What was the "retrieved piece"? Was it an unbiased selection, based on careful analysis and statistical evaluation of the distribution of the coils along the length of the cord, or was it any piece that appeared to have more coils than others?

Authors' Rebuttal: We thank the reviewer for highlighting the need to improve the clarity on this matter. To do so, we added an explanation to the methods section in which we define that the 'retrieved hUC piece' is the entire cord umbilical cord cut from the placenta leaving ~ 5-10 cm at the placenta. Therefore, we can confirm that the coiling index was calculated unbiased and without selection for a specific piece of the umbilical cord.

[Line 398-399: After birth, the hUCs were cut from the chorio-allantoic placenta (placenta) approximately 5-10 cm from the hUC insertion and...;

Line 406-407: The UCI was calculated by dividing the number of coils by the entire length of the retrieved hUC ...]

Reviewer's Response: The authors seem still not to have understood that the term "placenta" is short-hand for "chorio-allantoic placenta" and that the "chorion"ic component of the placenta is where the umbilical cord is inserted. Please use correct terminology. Kurt Benirschke, whom the authors cited as their justification in another answer, used outdated terminology that pre-dated the field's understanding of the development of the chorio-allantoic placenta.

- Is calculation of the UCI standardized across studies? If the standard method is to calculate the number of coils in just any retrieved piece without careful statistical analysis of the whereabouts of coils along the length, then the authors need to describe why this is acceptable rather than introduce a more reproducible and accurate method for measuring the UCI.

Authors' Rebuttal: The calculation of the UCI is indeed standardized across studies. It is standard practice to calculate the umbilical cord index based on the entire length of the umbilical cord. This is substantiated by the following references:

(A) Chan, J. S., & Baergen, R. N. (2012). Gross umbilical cord complications are associated with placental lesions of circulatory stasis and fetal hypoxia. *Pediatric and Developmental Pathology*, 15(6), 487-494.

(B) Benirschke K, Burton GJ, Baergen RN. *Pathology of the Human Placenta*. 6th ed. Berlin, Germany: Springer, 2012

(C) Dutman, A. C., & Nikkels, P. G. (2015). Umbilical hypercoiling in 2nd-and 3rdtrimester intrauterine fetal death. *Pediatric and Developmental Pathology*, 18(1), 10-16.

Reviewer's Response: These citations have not been included in the revised manuscript.

- Did the authors calculate the UCI before or after removal of the hUC from the chorion and, if so, is that standard?

Authors' Rebuttal: We calculate the umbilical cord index as a standard clinical procedure. We did so after removal from the placenta on the intact umbilical cord before any additional experiments. After removing the arteries from the Wharton's Jelly, the tissue was further processed. As the number helices of the umbilical cord do not change after removal from the placenta, the umbilical cord index can be assessed before and after separation.

Reviewer's Response: If the authors insist on using the incorrect "placenta" for "chorion", then at least place "chorion" in parentheses where relevant.

Cord length. The authors do not discuss the length of the cords, and the possibility that the hypercoiled cord, if stretched out and uncoiled, was longer than the normocoiled one. Perhaps hypercoiling is due to excessive growth? Given that excessive growth might have preceded hypercoiling, its molecular signature might not have been detected using the methods here.

Authors' Rebuttal: We appreciate the reviewer's insightful question and theory. We do not assume that hypercoiling is due to excessive growth, given Extensively Long Umbilical Cords (ELUCs) were associated with a variety of placental pathologies such as true knots but not with hypercoiling in a study conducted by Baergen et al.

Baergen, R. N., Malicki, D., Behling, C., & Benirschke, K. (2001). Morbidity, mortality, and placental pathology in excessively long umbilical cords: retrospective study. *Pediatric and Developmental Pathology*, 4(2), 144-153.

Reviewer's Response: The authors continue to ignore this alternative possibility in their interpretation of their results. If the authors insist, without any supporting data, that hyper-coiling is not due to the possibility of excessive growth, then they need to point this out and include the appropriate citation, above.

Cord length. The authors do not discuss the length of the cords, and the possibility that the hypercoiled cord, if stretched out and uncoiled, was longer than the normocoiled one. Perhaps hypercoiling is due to excessive growth? Given that excessive growth might have preceded hypercoiling, its molecular signature might not have been detected using the methods here.

Authors' Rebuttal: We appreciate the reviewer's insightful question and theory. We do not assume that hypercoiling is due to excessive growth, given Extensively Long Umbilical Cords (ELUCs) were associated with a variety of placental pathologies such as true knots but not with hypercoiling in a study conducted by Baergen et al.

Baergen, R. N., Malicki, D., Behling, C., & Benirschke, K. (2001). Morbidity, mortality, and placental pathology in excessively long umbilical cords: retrospective study. *Pediatric and Developmental Pathology*, 4(2), 144-153.

Additionally, our attention was directed towards examining the presence and molecular characteristics of hypercoiling. If hypercoiling were driven by excessive growth, we would expect finding differential expression of genes associated with growth or proliferation. Yet, such differential expression was not observed.

Reviewer's Response: Again, as pointed out, above, given that excessive growth might have preceded hyper-coiling, its molecular signature might not have been detected using methods here - the authors ignored this possibility as an alternative interpretation in their revision - .

Was there any difference in the uncoiled length of the hUC in the discordant twins? If there was a difference in cord length, this needs to be addressed as part of the interpretation of results, and in terms of sampling site, below.

Authors' Rebuttal: The length of the umbilical cord between the discordant twins was comparable.

[Line 144-146: The length of the hUC was comparable between the individuals of each of the four twin pairs and was within normal limits (30-70 cm).]

iBenirschke K, Burton GJ, Baergen RN. *Pathology of the Human Placenta*. 6th ed. Berlin, Germany: Springer, 2012

Reviewer's Response: This is still not clear - I asked for the uncoiled lengths, and the authors have ignored that request or explained why this is/is not relevant, and/or could not be calculated and, if the latter, the reported lengths do not represent the true lengths of the cords.

3. Sampling sites and physiology of the umbilical cord.

- Where along the length of the umbilical cords was sampling carried out? What were its dimensions and was the same piece used for all of the experiments? Or were different pieces used? The authors do not say; in the absence of this information, I can only assume that the authors used the same, possibly random piece for each hUC, thereby treating each cord's morphology and physiology as uniform - is it?

Authors' Rebuttal: We thank the reviewer for bringing attention to the requirement for further clarification. The reviewer is indeed correct, we used the same randomly selected piece of artery for the simultaneous isolation of DNA and RNA for all molecular measurements. The helical pattern was evenly spread throughout the umbilical cord, forming what is commonly referred to as a 'rope pattern'. Therefore, the sampling site is unbiased, and we can optimally link the methylomic and transcriptomic profiles. To improve clarity on this matter, we made some changes in the manuscript.

[Line 445-446: DNA and RNA were extracted from the same randomly selected snap-frozen artery fragment ...]

Reviewer's Response: "Random" and "unbiased" have specific meanings in statistical studies that have not been considered here - what do the authors mean by "random" and "unbiased", i.e., how did they select their pieces of hUC for sampling? Where did those pieces come from? Did sampling include a coil, was sampling between coils, did the pieces include a coil and the uncoiled region in between? Where along the length did the samples come from? Figure 1 shows a large amount of variation in the normo-coiled cords that the authors have not pointed out. Nor have the authors considered the possibility that they sampled in physiologically distinct sites - please see below.

Variations in the cord's physiology along its length would undoubtedly be reflected in gene expression. It would not be surprising if the differences observed between normo- and hypercoiled hUC could be attributable to normal physiological differences at the sampling sites if efforts were not made to match those sampling sites along the length of the cord. Sampling therefore should have been carried out at similar sites in each cord to account for the possibility of non-uniform physiology along its length, i.e., at some specific distance from its site of insertion into the chorion, at some specific distance from its site of insertion at the fetus, and/or perhaps at some fixed middle distance from either one of those sites.

Authors' Rebuttal: We appreciate this interesting thought raised by the reviewer. To our knowledge, the potential alteration in the molecular profile of the umbilical cord or umbilical cord arteries with increasing distance from the placenta has not been investigated. Since the selection of the umbilical cord segment for artery extraction was random, not within the first 5-10 cm of the connection to the placenta and coiling was uniformly distributed along the cord's length, we expect no interference with our comparative analysis.

Reviewer's Response: The authors have ignored the possibility that differential physiology along the length of the hUC's might explain the differences in molecular signature observed between the cords. If nothing is known about the status of the cord's physiology along its length, state it. Instead, they treated their "random" pieces as equivalent, without any statistical or scientific justification. Further, what is meant by "we expect no interference with our comparative analysis"?

Sampling site gets back to the issue of the distribution of the coils and the length of the cord, above, i.e., how comparable were the samples along the length of the cord, especially if hypercoils are not evenly distributed, or the cord lengths are dissimilar? These potential differences further argue for sampling at a set distance e.g., from the chorion, and/or fetus and, ideally, at multiple sites.

Authors' Rebuttal: Considering the uniform distribution of helices throughout the length of the umbilical cord as a rope pattern we also expect the samples along the cord to be highly comparative. [see Page 34: Figure 1: Overview of the included umbilical cords].

Reviewer's Response: On what basis can the authors "expect", i.e., assume, that sampling anywhere along hyper- and normo-coiled cords will be physiologically identical, as clearly, hyper-coiling is, as they have pointed out, not normal. Arterial hyper-coiling might be distinct from arterial physiology.

Thus, the authors' assumption that there will be no differences in molecular signatures along the length of hyper- or normo-coiled other than those due to hyper-coiling is a key assumption that needs to be highlighted and justified front and center in the Results, Discussion, and Methods. For if the authors believe that a normo-coiled and hyper-coiled umbilical cord are different only in arterial coiling, and that they can therefore sample anywhere along the length because all sites are physiologically equivalent, then they need to support that assumption from the outset, especially given the variation in the morphology of the normo-coiled cords.

What is the frequency of discordant cords in monozygotic twins? What is the frequency of concordant cords in monozygotic

twins, both normo- and hypercoiled?

Authors' Rebuttal: Studies investigating the frequency of umbilical cord coiling have predominantly focused on singleton pregnancies. To our knowledge, large epidemiological studies specifically examining uncomplicated monozygotic twins have not been conducted. Existing research suggests that umbilical cords in twin pairs tend to be slightly shorter and more coiled compared to singleton pregnancies (Edmonds HW. The spiral twist of the normal umbilical cord in twins and in singletons. *Am J Obstet Gynecol* 1954;67:102-120). However, it is important to note that twin pairs are often viewed as individual entities rather than directly comparing cord characteristics within the same pair.

Reviewer's Response: Given that the authors point out that existing research suggests that umbilical cords in twin pairs tend to be shorter and more coiled compared to singleton pregnancies, how valid, then, is the discordant twin comparison in this study? It would seem that even the normo-coiled cord is not entirely normal and could affect direct comparisons between the cords. Why isn't this pointed out? This needs to be highlighted and discussed.

Normocoiled monozygotic twins. I understand the authors' rationale for using discordant twins, claiming that the normocoiled hUC can be used as the control. However, they nevertheless need to include pairs of monozygotic twins each member of which exhibits normocoiled hUCs of similar lengths with sampling carried out in a standard manner as proscribed above, for the following reason: the authors need to show whether the prediction that there would be no meaningful differences between normocoiled twins is correct - if there are molecular differences, then the results of the present study, which apparently also show such differences (though with all of the aforementioned caveats regarding the samples used), may not be significant.

Authors' Rebuttal: We appreciate the reviewer's thoughtful consideration of our study. Our primary objective was to explore the origin of umbilical cord helices. By comparing hypercoiled umbilical cord arteries with normo-coiled arteries, we aimed to identify genes implicated in hypercoiling. Our findings revealed a subset of genes differentially expressed, which are involved in vascular development, cell-cell interaction, polarity, and axis formation, potentially contributing to the increased number of observed helices. We anticipate that umbilical cords with similar coiling indices would exhibit fewer transcriptomic differences, particularly for these genes. While we understand the suggestion to include pairs of monozygotic twins with identical coiling indices and lengths, it poses challenges in feasibility. It would require multiple pairs meeting such strict criteria, which may be impractical to obtain. Additionally, such an approach would essentially serve as a validation step for our findings. It would require a considerable allocation of resources without significantly enhancing our understanding. Therefore, we decided to not pursue the suggestion at this time.

Reviewer's Response: At the very least, this omission and its "impracticalities" need to be highlighted and discussed in the Discussion -

Results p. 8, line 175, section entitled "Hypercoiled umbilical arteries display a distinct transcriptomic profile": Shouldn't the authors have analysed 16 arteries, rather than 8, given that there are two arteries per cord? How did they distinguish members of each pair within a single hUC, or was each pair considered to be physiologically identical and what was the basis for that supposition?

Authors' Rebuttal: We appreciate the reviewer for bringing up this valid point. Indeed, each umbilical cord contains two umbilical cord arteries. Our decision to analyze only one artery per umbilical cord was based on the similarities between the two arteries. Both arteries typically exhibit the retained helical pattern. In our study, we aimed to unravel the mechanism underlying umbilical cord artery coiling. We expect the mechanism for coiling to be the same in both arteries and considered it sufficient to analyze one of the two arteries in this comparative investigation. However, we acknowledge that comparing both arteries may be relevant for future research questions, and we will consider this aspect in subsequent studies.

Reviewer #3 (Comments to the Authors (Required)):

The authors have done a thorough revision of the manuscript, addressing most of my concerns, and I do not have any further comments. I believe this study will be a valuable contribution to the field.

Ref#: LSA-2023-02543-TR

Manuscript – Twisting the theory on the origin of human umbilical cord coiling featuring monozygotic twins.

Dear dr. Eric Sawey,

Thank you for considering our manuscript and providing us with a second opportunity to resubmit our manuscript entitled ‘Twisting the theory on the origin of human umbilical cord coiling featuring monozygotic twins’. We were delighted to read that now both Reviewers 2 and 3 are enthusiastic about our work and view the manuscript as a valuable contribution to the field.

We obviously noted that Reviewer 1 remained critical. This was of course disappointing and unexpected since we specifically revised our manuscript to accommodate the criticism. In our revised manuscript, we provide ample support from scientific literature by undisputed top-experts for the terminology used in the manuscript and extensively substantiate the design and rationale of the study.

We again carefully studied the points made by Reviewer 1. We address each of them point-by-point response below and made further relevant changes to the manuscript.

We look forward to hearing your final decision on our manuscript.

Kind Regards,

On behalf of all authors,

Pia Todtenhaupt MSc

Lotte-Elisabeth van der Meeren MD PhD

Manuscript – Twisting the theory on the origin of human umbilical cord coiling featuring monozygotic twins.

Point-by-point response

Reviewer #1 (Comments to the Authors (Required)):

Reviewer Summary of the Authors' Rebuttal:

In this study, the authors conclude that some unknown property of the arteries is responsible for umbilical coiling. This conclusion is based on their having "randomly" sampled and compared the arterial vessels found within each of four sets of discordant twins, i.e., where one member of the pair exhibits normo-coiling and the other hyper-coiling, which is sometimes associated with fetal morbidity (though not here, as the authors have now mentioned).

The most important figure in this manuscript is Figure 1, which shows the umbilical cords of the four pairs of discordant twins. It forms the cornerstone of this study. What continues to confuse and has not yet been satisfactorily explained is how the authors "randomly" sampled the cords. Whilst the morphology of all four of the abnormal hyper-coiled cords looks generally uniform along their length, that of the normo-coiled cords is highly variable. The authors point out the former as their basis for "random" sampling but ignore the latter. This begs the question: What was sampled in these cords?

Authors' Response to the Reviewer's Response:

We conducted a random sampling process to ensure an unbiased selection of locations. In our previous revision, we highlighted the use of random sampling and clarified that the helical pattern was evenly distributed across the umbilical cords. This is also supported by Figure 1. We now additionally describe the exact steps taken during sampling of the umbilical cord.

[Line 436-439: To extract the hUC arteries, the collected hUC was cut into pieces of approximately 3 cm. All pieces were placed into a container with sterile PBS to wash off remaining blood. Thereafter, one hUC piece was randomly retrieved from the container and cut longitudinally to reveal the hUC arteries within the Wharton's Jelly.]

[Line 473-474: DNA and RNA were extracted from the same randomly selected snap-frozen artery fragment ...]

[Line 146: The helical pattern was evenly distributed throughout the hUC.]

The key point of our design is that any randomly sampled piece of cord the normo-coiled umbilical cord had fewer helices compared to any sampled location of the hypercoiled umbilical cord.

Manuscript – Twisting the theory on the origin of human umbilical cord coiling featuring monozygotic twins.

To highlight the importance of this omission, each of the distinct morphologies in the normo-coiled cords could have produced a different result when compared against the hypercoils. Thus, were the coils of the normo-coiled cords used? Or the non-coiled parts? If one morphological type was selected over the other, this would hardly constitute "random" sampling. Each morphology could have produced a different result; therefore, each type of morphology should have been tested independently. Then, if all the comparisons were identical, "random" sampling, presumably anywhere along the length of the cord (but why do I have to guess?), might be appropriate between the normo- and hyper-coiled cords.

Authors' Response to the Reviewer's Response:

The helical pattern was evenly distributed throughout all the umbilical cords [Figure 1A]. We acknowledge that the results may have been slightly different depending on the intensity of coiling of the sampled umbilical cords. Generally, we would expect a stronger difference in gene expression with increasing difference in umbilical cord coiling index. Therefore, if increased heterogeneity was present in the normocoiled umbilical cords, it would be less likely to detect a difference in gene expression rather than generating false positives. It would have been interesting to compare hypo-coiled and hyper-coiled umbilical cords from monochorionic twins to maximize the observed differences. However, we did not encounter such cases in our study population.

To accommodate the reviewers concern we have now added a section specifically describing the steps we took during sampling of the umbilical arteries.

[Line 436-439: *To extract the hUC arteries, the collected hUC was cut into pieces of approximately 3 cm. All pieces were placed into a container with sterile PBS to wash off remaining blood. Thereafter, one hUC piece was randomly retrieved from the container and cut longitudinally to reveal the hUC arteries within the Wharton's Jelly.*]

The authors' conclusions raise another issue that they did not address: if the arteries are responsible for creating umbilical coils, why are there so many fewer coils in the normo-coiled cords? After all, both cords have arteries running through them. The authors ignored the possibility of physiological variation in arterial function along the length of the cords, treating the cord as a uniformly inert pipe without any scientific justification. (Physiological variability would further argue against use of "random" sampling as the best approach for this study.)

Authors' Response to the Reviewer's Response:

It is indeed the case that each umbilical cord is build of the same components including that each umbilical cord has two arteries. Despite a uniform genetic background and components, there is undeniable phenotypic variation throughout the cord among them. We see a clear difference between the umbilical arteries upon removal [see figure below].

Manuscript – Twisting the theory on the origin of human umbilical cord coiling featuring monozygotic twins.

Figure 1B/C:

To accommodate the Reviewer's concern, we now elaborate on our hypothesis regarding the differences in coiling in the umbilical cord arteries in the discussion.

[Line 328-334: *The collagen fibers in the tunica media are arranged in a helical structure⁴¹. Muscle fiber alignment of a smooth muscle cell can vary with a maximum variation of 20 degrees⁴¹. This variance may be emphasized by the presence of a double muscle layer of which both layers are aligned in opposite directions resulting in a helix. The observed upregulation of type-I-collagen binding DCN and ASPN, stromal matrix component LUM and polarity associated CDH2 (N-cadherin) and LEFTY2 may facilitate structural adaptation in the artery, promoting its helical configuration.*]

Two more points. Firstly, the authors brush aside this Reviewer's suggestion that normal controls, i.e., concordant monozygotic twins, be included, claiming "impracticality". However difficult to achieve, they could nonetheless have addressed the strengths - or weaknesses - of this potentially critical control in the revised version.

Authors' Response to the Reviewer's Response:

In our manuscript we have presented the differences between hypercoiled and normocoiled umbilical arteries. The great strength of our study design is that by using monozygotic twins, we have both 'case' and 'control' at the same time. In our previous rebuttal, we provided reasoning for not analysing concordant monozygotic twins as a control. However, we do agree with the Reviewer, that future studies on the topic would ideally study a larger set of twins and more pieces per cord. Both would provide additional statistical and provide further insight in non-genetic biological variation within and between cords.

[Line 359-360: *Therefore, extending this comparison to a larger cohort may increase sensitivity to detect potential differential gene regulation in hUC arteries.*]

Manuscript – Twisting the theory on the origin of human umbilical cord coiling featuring monozygotic twins.

The primary reason was not impracticality; rather, we also believe that the addition of concordant monozygotic twin pairs would not significantly enhance our understanding of umbilical cord helices.

[Previous Rebuttal: We appreciate the reviewer's thoughtful consideration of our study. Our primary objective was to explore the origin of umbilical cord helices. By comparing hypercoiled umbilical cord arteries with normo-coiled arteries, we aimed to identify genes implicated in hypercoiling. Our findings revealed a subset of genes differentially expressed, which are involved in vascular development, cell-cell interaction, polarity, and axis formation, potentially contributing to the increased number of observed helices. We anticipate that umbilical cords with similar coiling indices would exhibit fewer transcriptomic differences, particularly for these genes. While we understand the suggestion to include pairs of monozygotic twins with identical coiling indices and lengths, it poses challenges in feasibility. It would require multiple pairs meeting such strict criteria, which may be impractical to obtain. Additionally, such an approach would essentially serve as a validation step for our findings. It would require a considerable allocation of resources without significantly enhancing our understanding. Therefore, we decided to not pursue the suggestion at this time.]

Secondly, it is further exasperating that the authors justify their incorrect use of the term "placenta" instead of "chorion" by citing usage of the former by Kurt Benirschke. Given what we have learned about the developmental biology of the placenta over the past several decades, that terminology is outdated and incorrect.

Authors' Response to the Reviewer's Response:

Reference Benirschke (Benirschke K, Burton GJ, Baergen RN. Pathology of the Human Placenta. 7th ed. Berlin, Germany: Springer, 2022) is a book that is regularly updated by experienced perinatal pathologists. The last edition is updated and published in 2022. This book is considered a solid reference by every perinatal pathologist across the world.

In our first revision, we have accommodated the reviewer's request to also refer to the term for the placenta of mammals; chorio-allantoic placenta. We are now additionally, specifying that the hUC is cut from the chorionic plate of the chorio-allantoic placenta.

[Line 423-424: After birth, the hUCs were cut from the chorionic plate of the chorio-allantoic placenta (placenta) approximately 5-10 cm from the hUC insertion ...]

Considering the extensive references that utilize the word *placenta* in humans with verifiable sources, it does not seem needed to refer to a specific location of the placenta (chorionic plate) or appropriate to use chorio-allantoic placenta throughout the manuscript. See recent reference:

1. Hivert, Marie-France, et al. "Placental IGFBP1 levels during early pregnancy and the risk of insulin resistance and gestational diabetes." *Nature Medicine* (2024): 1-7.
2. Powell, Theresa L., et al. "Synthesis of phospholipids in human placenta." *Placenta* 147 (2024): 12-20.

Manuscript – Twisting the theory on the origin of human umbilical cord coiling featuring monozygotic twins.

3. Wang, Meijiao, et al. "Single-nucleus multi-omic profiling of human placental syncytiotrophoblasts identifies cellular trajectories during pregnancy." Nature Genetics (2024): 1-12.

In summary, not only does the authors' lack of a detailed protocol regarding "random" sampling mean that their experiments can never be replicated by others, but their obfuscation means that readers cannot properly evaluate their data, leaving us to take them at their word. This is unacceptable.

Authors' Response to the Reviewer's Response:

To accommodate this, we have now detailed each step taken during the random sampling of the umbilical cord pieces in the material and methods section.

[Line 436-439: *To extract the hUC arteries, the collected hUC was cut into pieces of approximately 3 cm. All pieces were placed into a container with sterile PBS to wash off remaining blood. Thereafter, one hUC piece was randomly retrieved from the container and cut longitudinally to reveal the hUC arteries within the Wharton's Jelly.*]

Additionally, we would like to argue that performing random sampling increases the likelihood of replicability for other researchers, rather than diminishing it. Since we did not select specific segments from any of the four monozygotic twin pairs, we believe our results can be replicated with any part of the umbilical cord. Additionally, the protocol and raw data of the analysis are available upon request from the corresponding author.

Manuscript – Twisting the theory on the origin of human umbilical cord coiling featuring monozygotic twins.

Specific comments to the authors' rebuttal can be found below.

Authors' Rebuttal and Reviewer's Comments:

1. The coiling index and statistically evaluated distribution of coils along the length of the umbilical cord.

Reviewers initial comment: How exactly was the umbilical coiling index (UCI) calculated? The authors state in the Methods (p. 17, lines 402-403) that, "The UCI was calculated by dividing the number of coils by the length of the retrieved hUC piece as described by Strong et al." What was the "retrieved piece"? Was it an unbiased selection, based on careful analysis and statistical evaluation of the distribution of the coils along the length of the cord, or was it any piece that appeared to have more coils than others?

Authors' first rebuttal: We thank the reviewer for highlighting the need to improve the clarity on this matter. To do so, we added an explanation to the methods section in which we define that the 'retrieved hUC piece' is the entire cord umbilical cord cut from the placenta leaving ~ 5-10 cm at the placenta. Therefore, we can confirm that the coiling index was calculated unbiased and without selection for a specific piece of the umbilical cord.

[Line 398-399: After birth, the hUCs were cut from the chorio-allantoic placenta (placenta) approximately 5-10 cm from the hUC insertion and...;

Line 406-407: The UCI was calculated by dividing the number of coils by the entire length of the retrieved hUC ...]

Reviewer's Response: The authors seem still not to have understood that the term "placenta" is short-hand for "chorio-allantoic placenta" and that the "chorion"ic component of the placenta is where the umbilical cord is inserted. Please use correct terminology. Kurt Benirschke, whom the authors cited as their justification in another answer, used outdated terminology that predated the field's understanding of the development of the chorio-allantoic placenta.

Authors' Response to the Reviewer's Response:

We would like to refer to our initial rebuttal in which we mentioned that we acknowledge that this terminology chorio-allantoic placenta is correct, yet uncommonly used in practice.

[Previous rebuttal: *As the reviewer accurately pointed out, "chorio-allantoic placenta" is indeed the original developmental term for the mammalian placenta. While this formal terminology is technically correct, it is uncommonly used in practice. The shortened term "placenta" is much more prevalent in literature and books by renowned perinatal pathologists, such as the comprehensive work on placental pathology (Benirschke K, Burton GJ, Baergen RN. Pathology of the Human Placenta). To ensure clarity for a wide audience, we have opted to use the prevailing shortened vernacular "placenta". In response to the reviewer's valid*

Manuscript – Twisting the theory on the origin of human umbilical cord coiling featuring monozygotic twins.

observation, we have also added clarification in the methods section of the manuscript, explicitly stating that "placenta" refers to the "chorio-allantoic placenta".]

Additionally, reference Benirschke (Benirschke K, Burton GJ, Baergen RN. Pathology of the Human Placenta. 7th ed. Berlin, Germany: Springer, 2022) is a book that is regularly updated by experienced perinatal pathologists. The last edition is updated and published in 2022. This book is considered a solid reference by every perinatal pathologist across the world.

We are now additionally, specifying that the hUC is cut from the chorionic plate of the chorio-allantoic placenta.

[Line 423-424: After birth, the hUCs were cut from the chorionic plate of the chorio-allantoic placenta (placenta) approximately 5-10 cm from the hUC insertion ...]

Reviewers initial comment: Is calculation of the UCI standardized across studies? If the standard method is to calculate the number of coils in just any retrieved piece without careful statistical analysis of the whereabouts of coils along the length, then the authors need to describe why this is acceptable rather than introduce a more reproducible and accurate method for measuring the UCI.

Authors' first rebuttal: The calculation of the UCI is indeed standardized across studies. It is standard practice to calculate the umbilical cord index based on the entire length of the umbilical cord. This is substantiated by the following references:

- (A) Chan, J. S., & Baergen, R. N. (2012). Gross umbilical cord complications are associated with placental lesions of circulatory stasis and fetal hypoxia. *Pediatric and Developmental Pathology*, 15(6), 487-494.
- (B) Benirschke K, Burton GJ, Baergen RN. Pathology of the Human Placenta. 6th ed. Berlin, Germany: Springer, 2012
- (C) Dutman, A. C., & Nikkels, P. G. (2015). Umbilical hypercoiling in 2nd-and 3rdtrimester intrauterine fetal death. *Pediatric and Developmental Pathology*, 18(1), 10-16.

Reviewer's Response: These citations have not been included in the revised manuscript.

Authors' Response to the Reviewer's Response:

The given publications were indeed not added in the context of the umbilical cord index in our manuscript. We provided these to substantiate that this is standardized across studies. We have now updated the manuscript to also include these references [Line: 109].

Additionally, the UCI is calculated prior to sampling the artery across the entire length of the umbilical cord. The length of the umbilical cord, the number of helices, and the UCI are presented in Supplemental Table 1. We are confident that all relevant information is now included in the manuscript so that readers can exactly judge the procedures followed.

Manuscript – Twisting the theory on the origin of human umbilical cord coiling featuring monozygotic twins.

Reviewers initial comment: Did the authors calculate the UCI before or after removal of the hUC from the chorion and, if so, is that standard?

Authors' first rebuttal: We calculate the umbilical cord index as a standard clinical procedure. We did so after removal from the placenta on the intact umbilical cord before any additional experiments. After removing the arteries from the Wharton's Jelly, the tissue was further processed. As the number helices of the umbilical cord do not change after removal from the placenta, the umbilical cord index can be assessed before and after separation.

Reviewer's Response: If the authors insist on using the incorrect "placenta" for "chorion", then at least place "chorion" in parentheses where relevant.

Authors' Response to the Reviewer's Response:

Since we are not using the name chorion, we assume that the reviewer requests to place placenta in quotation marks and not chorion. Considering the extensive references that utilize the word placenta in humans without quotation marks in verifiable sources, we would argue that adding quotation marks may add to the confusion of the reader. See also following reference:

1. Benirschke K, Burton GJ, Baergen RN. Pathology of the Human Placenta. 7th ed. Berlin, Germany: Springer, 2022
2. Hivert, Marie-France, et al. "Placental IGFBP1 levels during early pregnancy and the risk of insulin resistance and gestational diabetes." *Nature Medicine* (2024): 1-7.
3. Powell, Theresa L., et al. "Synthesis of phospholipids in human placenta." Placenta 147 (2024): 12-20.
4. Wang, Meijiao, et al. "Single-nucleus multi-omic profiling of human placental syncytiotrophoblasts identifies cellular trajectories during pregnancy." *Nature Genetics* (2024): 1-12.

Reviewers initial comment: Cord length. The authors do not discuss the length of the cords, and the possibility that the hypercoiled cord, if stretched out and uncoiled, was longer than the normocoiled one. Perhaps hypercoiling is due to excessive growth? Given that excessive growth might have preceded hypercoiling, its molecular signature might not have been detected using the methods here.

Authors' fist rebuttal: We appreciate the reviewer's insightful question and theory. We do not assume that hypercoiling is due to excessive growth, given Extensively Long Umbilical Cords (ELUCs) were associated with a variety of placental pathologies such as true knots but not with hypercoiling in a study conducted by Baergen et al.

Baergen, R. N., Malicki, D., Behling, C., & Benirschke, K. (2001). Morbidity, mortality, and placental pathology in excessively long umbilical cords: retrospective study. *Pediatric and Developmental Pathology*, 4(2), 144-153.

Manuscript – Twisting the theory on the origin of human umbilical cord coiling featuring monozygotic twins.

Reviewer's Response: The authors continue to ignore this alternative possibility in their interpretation of their results. If the authors insist, without any supporting data, that hypercoiling is not due to the possibility of excessive growth, then they need to point this out and include the appropriate citation, above.

Authors' Response to the Reviewer's Response:

The theory that excessive growth underlies hypercoiling was an interesting idea which we had explored. We state in our manuscript that we did not observe a difference in length between the normo-coiled and hypercoiled umbilical cords.

[Line 144-146: The length of the hUC was comparable between the individuals of each of the four twin pairs and was within normal limits (30-70 cm)^{9,16}]

It still may be the case that in a fully uncoiled state the hypercoiled umbilical cords are longer than normocoiled cords. As uncoiling an umbilical cord is impossible, we are not able to provide the uncoiled length of the umbilical cord. If the increased number in coils was due to excessive growth, an extended length of the hypercoiled cord was expected. Yet, we provided in our previous rebuttal supporting data that extensively long umbilical cords were not associated with hypercoiling. We have now also added this information and the according reference to our manuscript.

[Line 253-255: *Additionally, excessive growth is unlikely to underlie hypercoiling, given Extensively Long Umbilical Cords (ELUC) were associated with a variety of placental pathologies such as true knots but not with hypercoiling in a previously conducted study*¹⁶.]

¹⁶ Baergen, R. N., Malicki, D., Behling, C., & Benirschke, K. (2001). Morbidity, mortality, and placental pathology in excessively long umbilical cords: retrospective study. *Pediatric and Developmental Pathology*, 4(2), 144-153.

Reviewers initial comment: Cord length. The authors do not discuss the length of the cords, and the possibility that the hypercoiled cord, if stretched out and uncoiled, was longer than the normocoiled one. Perhaps hypercoiling is due to excessive growth? Given that excessive growth might have preceded hypercoiling, its molecular signature might not have been detected using the methods here.

Authors' first rebuttal: We appreciate the reviewer's insightful question and theory. We do not assume that hypercoiling is due to excessive growth, given Extensively Long Umbilical Cords (ELUCs) were associated with a variety of placental pathologies such as true knots but not with hypercoiling in a study conducted by Baergen et al.

Baergen, R. N., Malicki, D., Behling, C., & Benirschke, K. (2001). Morbidity, mortality, and placental pathology in excessively long umbilical cords: retrospective study. *Pediatric and Developmental Pathology*, 4(2), 144-153.

Manuscript – Twisting the theory on the origin of human umbilical cord coiling featuring monozygotic twins.

Additionally, our attention was directed towards examining the presence and molecular characteristics of hypercoiling. If hypercoiling were driven by excessive growth, we would expect finding differential expression of genes associated with growth or proliferation. Yet, such differential expression was not observed.

Reviewer's Response: Again, as pointed out, above, given that excessive growth might have preceded hyper-coiling, its molecular signature might not have been detected using methods here - the authors ignored this possibility as an alternative interpretation in their revision - .

Authors' Response to the Reviewer's Response:

If excessive growth would underlie the increased number in helices, we would expect hypercoiled cords to be of increased length. In our manuscript, we mention that the length between the normo-coiled and hypercoiled umbilical cords were comparable. Additionally, extensively long umbilical cords were in previous studies not associated with hypercoiling. We have now added this information to the discussion section of our manuscript. On top of this, for more transparency, we had added the length of the umbilical cords to supplementary table 1.

[Line 253-255: *Additionally, excessive growth is unlikely to underlie hypercoiling, given Extensively Long Umbilical Cords (ELUC) were associated with a variety of placental pathologies such as true knots but not with hypercoiling in a previously conducted study*¹⁶.]

¹⁶ Baergen, R. N., Malicki, D., Behling, C., & Benirschke, K. (2001). Morbidity, mortality, and placental pathology in excessively long umbilical cords: retrospective study. *Pediatric and Developmental Pathology*, 4(2), 144-153.

[Supplemental Table 1: Donor and umbilical cord characteristics of monozygotic twin pairs discordant for coiling]

Reviewers initial comment: Was there any difference in the uncoiled length of the hUC in the discordant twins? If there was a difference in cord length, this needs to be addressed as part of the interpretation of results, and in terms of sampling site, below.

Authors' first rebuttal: The length of the umbilical cord between the discordant twins was comparable.

[Line 144-146: The length of the hUC was comparable between the individuals of each of the four twin pairs and was within normal limits (30-70 cm)ⁱ.]

ⁱBenirschke K, Burton GJ, Baergen RN. *Pathology of the Human Placenta*. 6th ed. Berlin, Germany: Springer, 2012

Manuscript – Twisting the theory on the origin of human umbilical cord coiling featuring monozygotic twins.

Reviewer's Response: This is still not clear - I asked for the uncoiled lengths, and the authors have ignored that request or explained why this is/is not relevant, and/or could not be calculated and, if the latter, the reported lengths do not represent the true lengths of the cords.

Authors' Response to the Reviewer's Response:

The length of the umbilical cord we present in supplemental table 1 is the length after removal from the placenta after birth. Since the umbilical cord cannot be stretched to a point where the helices are unwound, it is not feasible to determine its uncoiled length. As far as we know, no publications provide the uncoiled length of the umbilical cord or describe methods to measure it. Consequently, we argue that the measured length of the cord, as presented in our data, represents its 'true' length. We agree that if uncoiling had been possible, the result may be that an uncoiled hypercoiled umbilical cord is longer than the cord was in its original hypercoiled state. That said, extensively long umbilical cords were previously not associated with hypercoiling indicating that cord growth and cord coiling are independent processes.

Baergen, R. N., Malicki, D., Behling, C., & Benirschke, K. (2001). Morbidity, mortality, and placental pathology in excessively long umbilical cords: retrospective study. *Pediatric and Developmental Pathology*, 4(2), 144-153.

3. Sampling sites and physiology of the umbilical cord.

Reviewers initial comment: Where along the length of the umbilical cords was sampling carried out? What were its dimensions and was the same piece used for all of the experiments? Or were different pieces used? The authors do not say; in the absence of this information, I can only assume that the authors used the same, possibly random piece for each hUC, thereby treating each cord's morphology and physiology as uniform - is it?

Authors' first rebuttal: We thank the reviewer for bringing attention to the requirement for further clarification. The reviewer is indeed correct, we used the same randomly selected piece of artery for the simultaneous isolation of DNA and RNA for all molecular measurements. The helical pattern was evenly spread throughout the umbilical cord, forming what is commonly referred to as a 'rope pattern'. Therefore, the sampling site is unbiased, and we can optimally link the methylomic and transcriptomic profiles. To improve clarity on this matter, we made some changes in the manuscript.

[Line 445-446: DNA and RNA were extracted from the same randomly selected snap-frozen artery fragment ...]

Reviewer's Response: "Random" and "unbiased" have specific meanings in statistical studies that have not been considered here - what do the authors mean by "random" and "unbiased", i.e., how did they select their pieces of hUC for sampling? Where did those pieces come from? Did sampling include a coil, was sampling between coils, did the pieces include a coil and the uncoiled region in between? Where along the length did the samples come from? Figure 1

Manuscript – Twisting the theory on the origin of human umbilical cord coiling featuring monozygotic twins.
shows a large amount of variation in the normo-coiled cords that the authors have not pointed out. Nor have the authors considered the possibility that they sampled in physiologically distinct sites - please see below.

Authors' Response to the Reviewer's Response:

We used a random sampling process to ensure that our selection of locations was unbiased. In our prior revision, we emphasized the use of random sampling and explained that the helical pattern was evenly spread throughout the umbilical cords. Figure 1 provides additional evidence to support this. The key point of our design is that any randomly sampled piece of cord the normo-coiled umbilical cord had fewer helices compared to any sampled location of the hypercoiled umbilical cord. To further improve clarification on the sampling method, we now explain each step in the methods section of our manuscript.

[Line 436-439: *To extract the hUC arteries, the collected hUC was cut into pieces of approximately 3 cm. All pieces were placed into a container with sterile PBS to wash off remaining blood. Thereafter, one hUC piece was randomly retrieved from the container and cut longitudinally to reveal the hUC arteries within the Wharton's Jelly.*]

[Line 473-474: *DNA and RNA were extracted from the same randomly selected snap-frozen artery fragment ...*]

[Line 146: *The helical pattern was evenly distributed throughout the hUC.*]

Reviewers initial comment: Variations in the cord's physiology along its length would undoubtedly be reflected in gene expression. It would not be surprising if the differences observed between normo- and hypercoiled hUC could be attributable to normal physiological differences at the sampling sites if efforts were not made to match those sampling sites along the length of the cord. Sampling therefore should have been carried out at similar sites in each cord to account for the possibility of non-uniform physiology along its length, i.e., at some specific distance from its site of insertion into the chorion, at some specific distance from its site of insertion at the fetus, and/or perhaps at some fixed middle distance from either one of those sites.

Authors' first rebuttal: We appreciate this interesting thought raised by the reviewer. To our knowledge, the potential alteration in the molecular profile of the umbilical cord or umbilical cord arteries with increasing distance from the placenta has not been investigated. Since the selection of the umbilical cord segment for artery extraction was random, not within the first 5-10 cm of the connection to the placenta and coiling was uniformly distributed along the cord's length, we expect no interference with our comparative analysis.

Reviewer's Response: The authors have ignored the possibility that differential physiology along the length of the hUC's might explain the differences in molecular signature observed between the cords. If nothing is known about the status of the cord's physiology along its length, state it. Instead, they treated their "random" pieces as equivalent, without any statistical or

Manuscript – Twisting the theory on the origin of human umbilical cord coiling featuring monozygotic twins. scientific justification. Further, what is meant by "we expect no interference with our comparative analysis"?

Authors' Response to the Reviewer's Response:

We do not expect interference with our comparative analysis as no specific location in the umbilical cord was favoured over others making the comparison between normo- and hypercoiled umbilical cords robust across locations. We recognize that the results might vary slightly depending on the level of coiling of the sampled umbilical cords. Typically, we would anticipate a greater variation in gene expression with larger differences in the umbilical cord coiling index. However, there is still a clear difference between the two sampled umbilical cord pieces, regardless of where they were taken from.

Reviewers initial comment: Sampling site gets back to the issue of the distribution of the coils and the length of the cord, above, i.e., how comparable were the samples along the length of the cord, especially if hypercoils are not evenly distributed, or the cord lengths are dissimilar? These potential differences further argue for sampling at a set distance e.g., from the chorion, and/or fetus and, ideally, at multiple sites.

Authors' first rebuttal: Considering the uniform distribution of helices throughout the length of the umbilical cord as a rope pattern we also expect the samples along the cord to be highly comparative. [see Page 34: Figure 1: Overview of the included umbilical cords].

Reviewer's Response: On what basis can the authors "expect", i.e., assume, that sampling anywhere along hyper- and normo-coiled cords will be physiologically identical, as clearly, hyper-coiling is, as they have pointed out, not normal. Arterial hyper-coiling might be distinct from arterial physiology.

Thus, the authors' assumption that there will be no differences in molecular signatures along the length of hyper- or normo-coiled other than those due to hyper-coiling is a key assumption that needs to be highlighted and justified front and center in the Results, Discussion, and Methods. For if the authors believe that a normo-coiled and hyper-coiled umbilical cord are different only in arterial coiling, and that they can therefore sample anywhere along the length because all sites are physiologically equivalent, then they need to support that assumption from the outset, especially given the variation in the morphology of the normo-coiled cords.

Authors' Response to the Reviewer's Response:

The critical aspect of our comparison is contrasting a hypercoiled umbilical cord with a normo-coiled one within the same genetic background. We do not assume that there is no difference in molecular signature along the length of the cord; however, we believe that using random sampling to compare these pieces enhances the robustness of our results, suggesting that the observed differences are consistent regardless of the samples' location.

Manuscript – Twisting the theory on the origin of human umbilical cord coiling featuring monozygotic twins.

Reviewers initial comment: What is the frequency of discordant cords in monozygotic twins? What is the frequency of concordant cords in monozygotic twins, both normo- and hypercoiled?

Authors' first rebuttal: Studies investigating the frequency of umbilical cord coiling have predominantly focused on singleton pregnancies. To our knowledge, large epidemiological studies specifically examining uncomplicated monozygotic twins have not been conducted. Existing research suggests that umbilical cords in twin pairs tend to be slightly shorter and more coiled compared to singleton pregnancies (Edmonds HW. The spiral twist of the normal umbilical cord in twins and in singletons. Am J Obstet Gynecol 1954;67:102-120). However, it is important to note that twin pairs are often viewed as individual entities rather than directly comparing cord characteristics within the same pair.

Reviewer's Response: Given that the authors point out that existing research suggests that umbilical cords in twin pairs tend to be shorter and more coiled compared to singleton pregnancies, how valid, then, is the discordant twin comparison in this study? It would seem that even the normo-coiled cord is not entirely normal and could affect direct comparisons between the cords. Why isn't this pointed out? This needs to be highlighted and discussed.

Authors' Response to the Reviewer's Response:

In this study we are investigating the difference and origin of umbilical cord coiling. For this, coiling discordant monozygotic twin pairs are as a valuable model to exclude genetic origin, many environmental factors, and parental origins. Although umbilical cords of twin pairs as a group tend to be slightly shorter and more coiled as compared with singletons, this does not affect the contrast we study here, that is between normo-coiled and hyper-coiled because these groups are based on accepted cut-off values of the coiling index used for both singletons and twins.

Reviewers initial comment: Normocoiled monozygotic twins. I understand the authors' rationale for using discordant twins, claiming that the normocoiled hUC can be used as the control. However, they nevertheless need to include pairs of monozygotic twins each member of which exhibits normocoiled hUCs of similar lengths with sampling carried out in a standard manner as proscribed above, for the following reason: the authors need to show whether the prediction that there would be no meaningful differences between normocoiled twins is correct - if there are molecular differences, then the results of the present study, which apparently also show such differences (though with all of the aforementioned caveats regarding the samples used), may not be significant.

Authors' first rebuttal: We appreciate the reviewer's thoughtful consideration of our study. Our primary objective was to explore the origin of umbilical cord helices. By comparing hypercoiled umbilical cord arteries with normo-coiled arteries, we aimed to identify genes

Manuscript – Twisting the theory on the origin of human umbilical cord coiling featuring monozygotic twins.

implicated in hypercoiling. Our findings revealed a subset of genes differentially expressed, which are involved in vascular development, cell-cell interaction, polarity, and axis formation, potentially contributing to the increased number of observed helices. We anticipate that umbilical cords with similar coiling indices would exhibit fewer transcriptomic differences, particularly for these genes. While we understand the suggestion to include pairs of monozygotic twins with identical coiling indices and lengths, it poses challenges in feasibility. It would require multiple pairs meeting such strict criteria, which may be impractical to obtain. Additionally, such an approach would essentially serve as a validation step for our findings. It would require a considerable allocation of resources without significantly enhancing our understanding. Therefore, we decided to not pursue the suggestion at this time.

Reviewer's Response: At the very least, this omission and its "impracticalities" need to be highlighted and discussed in the Discussion -

Authors' Response to the Reviewer's Response:

The key strength of our study design is that by using monozygotic twins, we effectively have both 'case' and 'control' simultaneously. In our previous rebuttal, we thoroughly explained our choice not to use concordant monozygotic twins as a control group. The main reason was not impracticality but that we believe that adding concordant monozygotic twin pairs would not substantially improve our understanding of umbilical cord helices. However, we do agree with the Reviewer, that future studies on the topic would ideally study a larger set of twins and more pieces per cord. Both would provide additional statistical power and thereby further insight in non-genetic biological variation within and between cords.

Reviewers initial comment: Results p. 8, line 175, section entitled "Hypercoiled umbilical arteries display a distinct transcriptomic profile": Shouldn't the authors have analysed 16 arteries, rather than 8, given that there are two arteries per cord? How did they distinguish members of each pair within a single hUC, or was each pair considered to be physiologically identical and what was the basis for that supposition?

Authors' first rebuttal: We appreciate the reviewer for bringing up this valid point. Indeed, each umbilical cord contains two umbilical cord arteries. Our decision to analyze only one artery per umbilical cord was based on the similarities between the two arteries. Both arteries typically exhibit the retained helical pattern. In our study, we aimed to unravel the mechanism underlying umbilical cord artery coiling. We expect the mechanism for coiling to be the same in both arteries and considered it sufficient to analyze one of the two arteries in this comparative investigation. However, we acknowledge that comparing both arteries may be relevant for future research questions, and we will consider this aspect in subsequent studies.

Manuscript – Twisting the theory on the origin of human umbilical cord coiling featuring monozygotic twins.

Reviewer #3 (Comments to the Authors (Required)):

The authors have done a thorough revision of the manuscript, addressing most of my concerns, and I do not have any further comments. I believe this study will be a valuable contribution to the field.

Authors' Response to the Reviewer's Response:

We are glad to read that we were able to address the Reviewer's concerns, and we appreciate the support for our study. We are delighted to hear that our work is seen as a valuable contribution to the field.

May 13, 2024

RE: Life Science Alliance Manuscript #LSA-2023-02543-TRR

Dr. Lotte E. van der Meeren
Leiden University Medical Center
Department of Pathology
Eindhovenweg 20
Leiden 2333 ZC Leiden
Netherlands

Dear Dr. van der Meeren,

Thank you for submitting your revised manuscript entitled "Twisting the theory on the origin of human umbilical cord coiling featuring monozygotic twins". We would be happy to publish your paper in Life Science Alliance pending final revisions necessary to meet our formatting guidelines.

- please be sure that the authorship listing and order is correct
- please add ORCID ID for the corresponding and secondary author -- you should have received instructions on how to do so
- please add the Twitter handle of your host institute/organization as well as your own or/and one of the authors in our system
- please remove the Graphical Abstract from the manuscript file and leave it uploaded separately
- please remove track changes from the manuscript file
- please incorporate any points from the Conclusion section into the Discussion; we only allow a Discussion section
- please add your main, supplementary figure, and table legends to the main manuscript text after the references section
- please add callouts for Figures 4A, B and S3A, B to your main manuscript text

LSA now encourages authors to provide a 30-60 second video where the study is briefly explained. We will use these videos on social media to promote the published paper and the presenting author (for examples, see <https://docs.google.com/document/d/1-UWCfbE4pGcDdcgzcmiuJI2XMBJnxKYeqRvLLrLS08s/edit?usp=sharing>). Corresponding or first-authors are welcome to submit the video. Please submit only one video per manuscript. The video can be emailed to contact@life-science-alliance.org

A. FINAL FILES:

B. MANUSCRIPT ORGANIZATION AND FORMATTING:

Sincerely,

May 22, 2024

RE: Life Science Alliance Manuscript #LSA-2023-02543-TRRR

Lotte Elisabeth van der Meeren
Leiden University Medical Center

Dear Dr. van der Meeren,

Thank you for submitting your Resource entitled "Twisting the theory on the origin of human umbilical cord coiling featuring monozygotic twins". It is a pleasure to let you know that your manuscript is now accepted for publication in Life Science Alliance. Congratulations on this interesting work.

DISTRIBUTION OF MATERIALS:

Again, congratulations on a very nice paper. I hope you found the review process to be constructive and are pleased with how the manuscript was handled editorially. We look forward to future exciting submissions from your lab.

Sincerely,
